# An updated floristic map of the world

Yunpeng Liu [1,2,11], Xiaoting Xu [1,3,11], Dimitar Dimitrov [4,11], Loic Pellissier [5,6], Michael K. Borregaard [2], Nawal Shrestha[1,7], Xiangyan Su[1,8], Ao Luo[1], Niklaus E. Zimmermann [6], Carsten Rahbek [1,2,9,10] ✉ & Zhiheng Wang [1] ✉

Floristic regions reflect the geographic organization of floras and provide essential tools for biological studies. Previous global floristic regions are generally based on floristic endemism, lacking a phylogenetic consideration that captures floristic evolution. Moreover, the contribution of tectonic dynamics and historical and current climate to the division of floristic regions remains unknown. Here, by integrating global distributions and a phylogeny of 12,664 angiosperm genera, we update global floristic regions and explore their temporal changes. Eight floristic realms and 16 nested sub-realms are identified. The previously-defined Holarctic, Neotropical and Australian realms are recognized, but Paleotropical, Antarctic and Cape realms are not. Most realms have formed since Paleogene. Geographic isolation induced by plate tectonics dominates the formation of floristic realms, while current/historical climate has little contribution. Our study demonstrates the necessity of integrating distributions and phylogenies in regionalizing floristic realms and the interplay of macroevolutionary and paleogeographic processes in shaping regional floras.

Biogeographic regionalizations divide the world into regions based on the similarity between faunas or floras and have been one of the central topics in biogeography since the time of Charles Darwin and Alfred Russel Wallace[1–4]. Darwin's biogeographic observations on the similarities in faunas or floras across regions during the voyage of Beagle (1831–1836) led him to his theory on natural selection[3,5]. In 1876, Wallace published his global map of zoological realms based on compositional similarity and taxonomic relationships of animal families across regions. The early maps of biogeographic regions, including the Wallace zoological realms, have significantly improved our understanding of global biodiversity[6], and provided a spatially explicit tool for conservation planning[7,8].

To understand the evolution of plant diversity, several global floristic regionalization schemes were generated by early authors, including de Candolle (1820)[9,10], Schouw (1823)[4], Engler (1892) and others[1,4,10–12]. Later, Takhtajan (1969, 1970, 1974, 1978, and 1986) summarized the basic understanding on the distribution and origin of floras and developed the most widely used map of floristic biogeographic regionalization scheme up to now, which divided the world landmasses into six "kingdoms" and 35 "regions"[4]. These early floristic schemes identified biogeographic regions and their hierarchical relationships mainly based on endemism of floras at different taxonomic levels, combined with a generally descriptive understanding of paleoclimate and geological history[1,4,10–12]. The rapid accumulation of

[1]Institute of Ecology, College of Urban and Environmental Sciences, and Key Laboratory of Earth Surface Processes of Ministry of Education, Peking University, 100871 Beijing, China. [2]Center for Macroecology, Evolution and Climate, Natural History Museum of Denmark, University of Copenhagen, Universitetsparken 15, 2100 Copenhagen Ø, Denmark. [3]Key Laboratory of Bio-Resource and Eco-Environment of Ministry of Education, College of Life Sciences, Sichuan University, 610065 Chengdu, Sichuan, China. [4]Department of Natural History, University Museum of Bergen, University of Bergen, Postbox 7800, 5020 Bergen, Norway. [5]Landscape Ecology, Institute of Terrestrial Ecosystems, ETH Zurich, 8092 Zurich, Switzerland. [6]Swiss Federal Research Institute WSL, 8903 Birmensdorf, Switzerland. [7]State Key Laboratory of Grassland Agro-ecosystems, Institute of Innovation Ecology, Lanzhou University, 730000 Lanzhou, China. [8]Land Consolidations and Rehabilitation Center, Ministry of Natural Resources, 100035 Beijing, China. [9]Center for Global Mountain Biodiversity, GLOBE Institute, University of Copenhagen, Universitetsparken 15, 2100 Copenhagen, Denmark. [10]Danish Institute for Advanced Study, University of Southern Denmark, 5230 Odense M, Denmark. [11]These authors contributed equally: Yunpeng Liu, Xiaoting Xu, Dimitar Dimitrov. ✉e-mail: crahbek@snm.ku.dk; zhiheng.wang@pku.edu.cn

phylogenetic and species distribution data has significantly improved the development of quantitative and repeatable biogeographic regionalizations, providing valuable insight on the historical relationships among floras or faunas[2,13–19]. For example, an updated global zoogeographic regionalization[2] was generated recently using quantitative phylogenetic relatedness of birds, mammals and amphibians showing the geographic variations in tetrapod evolution at the global scale[20,21]. A few regional floristic regionalizations using similar methods have been conducted in China[17,22], Japan[23], South Africa[15], and the tropics[14]. However, the progress in building a global floristic regionalization using quantitative approaches has lagged behind state-of-the-art zoogeographic regionalizations due to the lack of phylogenetic and distribution data of plants at a global scale. Recently, Carta et al. proposed a global floristic regionalization with three floristic kingdoms based on phylogenetic beta diversity of a fraction (ca. 20%) of vascular plant species[24]. The impacts of incomplete sampling in both distribution and phylogeny on the regionalization of Carta et al.[24] remain unknown. Meanwhile, this regionalization was based on species rather than genera, limiting its comparison with previous floristic regionalizations based on genus and family endemism[4,11,12].

The drivers shaping boundaries between different biogeographic regions are critical for understanding the macroevolution of floras and faunas in different regions[20], yet they remain rarely explored[20,25]. The dynamics in geology and macroclimate over time have left profound effects on the speciation, extinction, and dispersal of plants and thus are considered as drivers shaping floristic boundaries[20,21,26,27]. The dynamics in plate tectonics throughout the earth's history, including plate collision, orogeny and the emergence and breakup of land bridges, have led to significant changes in geographic isolation and floristic exchanges among landmasses over time, subsequently influencing evolutionary processes such as speciation and extinction of floras in different regions[20,28,29]. Macroclimate, especially temperature and precipitation, represents a major dimension of the ecological niches of plants[30,31]. Geographic differences in paleoclimate during geological times led to significant changes in species compositions across space because of the effect of climatic filtering on species distributions[30,32]. A recent study indicated that current climate and tectonic movements contributed to the boundaries between zoological realms[20]. However, the spatiotemporal variation in the relative roles of these two drivers on the boundaries between global floristic realms remain unknown. Moreover, isolation-induced clade splitting, and the independent radiation of descendant lineages have led to abrupt floristic transitions across biogeographic boundaries[33]. In addition, different clades may contribute unequally to the division of different realms due to differences in clade evolutionary histories and geographic distributions[34], which remains to be evaluated.

Here, we present a global map of floristic realms by integrating distribution data and a phylogeny of 12,664 angiosperm genera (ca. 85% of all known angiosperm genera). Floristic realms are identified using hierarchical clustering methods based on phylogenetic beta diversity between regions at genus level. We then demonstrate the temporal dynamics of the identified floristic realms during the Cretaceous and the Cenozoic, and further compare our work with Takhtajan's floristic map[4] and Holt et al.'s zoological map[2]. To explore the mechanisms underlying the formation of the identified floristic realms, we evaluate the effects of contemporary climate, the dynamics of geographic isolation induced by long-term plate tectonics, and historical climate. We also evaluate the relative contributions of clade splitting events at different geological times to realm divisions.

## Results and discussion
### World's floristic regionalization

Hierarchical clustering analysis based on phylogenetic beta diversity (see Methods) indicates that the terrestrial world is divided into eight floristic realms, namely African, Australian, Novozealandic, Indo-Malesian, Neotropical, Chile-Patagonian, Holarctic and Saharo-Arabian realms (Fig. 1a). The African realm is closely related to the Indo-Malesian realm, the Australian realm is closely related to the Novozealandic realm, and the Neotropical realm is closely related to the Chile-Patagonian realm (Fig. 1b). The above realms are grouped into

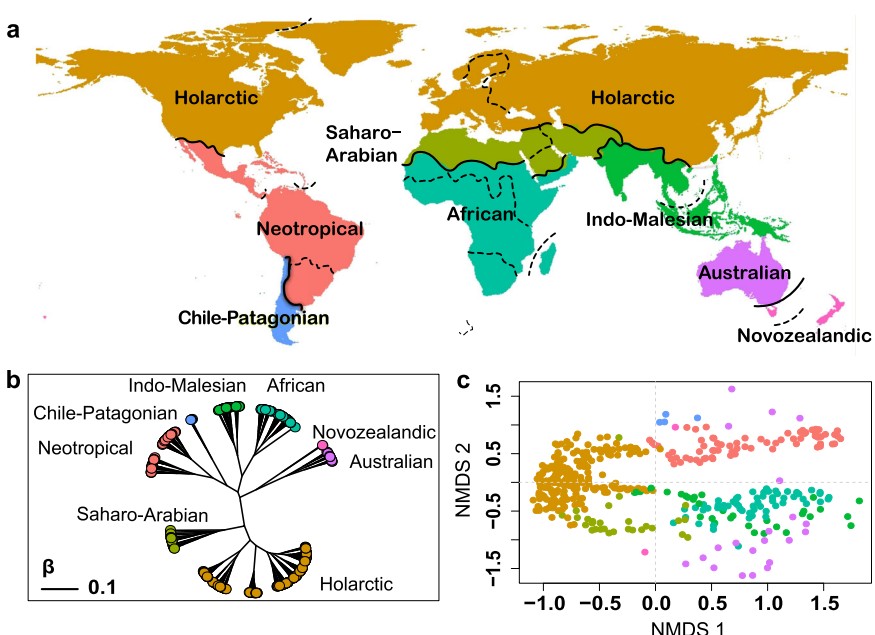

**Fig. 1 | Floristic realms and sub-realms of the world. a** Boundaries of the eight floristic realms and 16 sub-realms are shown in solid and dashed lines, respectively. **b** The unrooted dendrogram depicts the relationships among floristic realms evaluated using UPGMA clustering method based on phylogenetic beta diversity between realms. The scale bar in the dendrogram shows the dissimilarity between realms. **c** The scatter plot shows the dissimilarities in the phylogenetic compositions between different geographic standard units (GSU) generated using non-metric multidimensional scaling (NMDS) ordination. Each tip in the dendrogram and each point in the scatter plot represents a geographic standard unit and the colors indicate the floristic realms that they belong to. For comparison, the floristic realms and sub-realms based on trees with alternative dating constraints are shown in Supplementary Fig. 2. Source data are provided as a Source Data file.

the Gondwanan super-realm. The Holarctic and Saharo-Arabian realms are grouped into the Laurasian super-realm. Within these eight realms, we further identified 16 floristic sub-realms (Fig. 1; also see Supplementary Table 1 for their names and Supplementary Fig. 1 for their relationships).

The boundaries between different realms generally have high confidence (i.e., "hardness") in most cases as evaluated by silhouette analysis[35] (Supplementary Fig. 2, see Methods) and are robust to different assumptions on the crown age of angiosperms (Supplementary Fig. 3) and to variations in phylogenetic topology (Supplementary Figs. 4–8). We also evaluated the effect of incomplete sampling on realm boundaries by repeating our regionalization process using only the genera included in ref. 24 (9905 angiosperm genera), and we found that the boundaries were also robust (Supplementary Fig. 9, also see Supplementary Discussions for details). Interestingly, distribution ranges for ca. 53.6% genera do not extend beyond the identified boundaries (Supplementary Data 5), suggesting that the identified realm boundaries may reflect ecological or evolutionary barriers of plant distributions[31]. Although the boundaries identified by the UPGMA clustering are mostly consistent with those based on the fuzzy clustering method (i.e., fuzzy c-means), inconsistency exits in the identification of the Chile-Patagonian realm and the North American sub-realm, suggesting lower confidence in the identification of these realms/sub-realms than others (see Supplementary Fig. 17, Methods, and Supplementary Discussions for details).

Floristic realms and their relatedness based on taxonomic beta diversity are relatively consistent with phylogenetic-based floristic realms with three exceptions (Supplementary Fig. 10, also see Supplementary results and discussion for a detailed comparison). First, subtropical East Asia was grouped into the Indo-Malaysian realm when taxonomic beta diversity was used but was grouped into the Holarctic realm when phylogenetic beta diversity was used. Second, Mexico was grouped into the Chile-Patagonian realm when taxonomic beta diversity was used but was grouped into the Neotropical realm when phylogenetic beta diversity was used. Third, (Australian, Chile-Patagonian) realms are grouped with (African, Indo-Malesian) realms, but was groped with ((African, Indo-Malesian), (Neotropical, Chile-Patagonian)) realms when phylogenetic beta diversity was used. These differences between the taxonomic-based and phylogenetic-based results may suggest recent exchange through dispersal of lineages among these regions (see Supplementary Discussion for details).

## The divergent times between the identified realms

By cutting the dated phylogeny at different depths (i.e., geological times), we found that the identified floristic realms have not become distinct before the Cretaceous (160 Ma; Fig. 2 & Fig. 3). During the Early Cretaceous, the divergences in present-day floras in the Gondwanan and Laurasian super-realms had formed (Fig. 2c). The divergences between the floras of Neotropical and African realms were not clear until about 80 Ma ago. During the Cenozoic, the dissimilarity between the present-day floras of different realms significantly increased (Fig. 3).

The divergences among the present-day floras of most realms have formed since the early Cenozoic, and the boundaries between them remain largely unchanged towards the present with two exceptions (Fig. 2). First, the boundaries between the African and the Indo-Malesian realms disappeared when evaluated at the phylogenetic depth from the Eocene to the Pliocene, although their divergence was clear during the Paleogene (60 Ma; Fig. 2). The dissimilarity between the present-day floras of these two realms was much lower at the phylogenetic depth from the Eocene to the Pliocene than at the present (0 Ma), resulting in the disappearance of boundaries of these two realms at these times (Fig. 3). Fossil evidence on historical changes in woody assemblages during the Cenozoic[36] further supports this finding, which is possibly because the northward drifted of the Indian

subcontinent during the Cenozoic accumulated floristic exchanges between Eurasia and Africa[37,38].

Second, the northern boundary of the Neotropics realm ends at the Greater Antilles and the Yucatan Peninsula in Mesoamerica during 60–40 Ma and further extends to Mexico afterwards (Fig. 2). These results are consistent with recent findings about the history of biotic interchange between South America and Mesoamerica[39–42]. Recent studies[39,40] found that dispersal of plant lineages from Amazonia to Mesoamerica and the Caribbean islands occurred continuously since the early Cenozoic and was much more frequent than dispersal from Amazonia to any adjacent regions in the south. Fossil and plate tectonic evidence suggest that the expansion of megathermal vegetation[43] and an emergent Aves Ridge during the Paleocene[42,44] may have facilitated the biotic interchange from South America to Mesoamerica and the Caribbean islands, which may have led to high phylogenetic similarities of flora in these regions.

It is noteworthy that the inference of the divergence times between the identified realms was based on the phylogeny of present-day taxa. Although these results are relatively consistent with fossil evidence, how to reconstruct historical divergences between floristic realms by integrating distributions of current clades and fossils remains to be explored in future studies. Such studies may need a comprehensive framework that integrates analytical tools in paleoecology, systematics, paleoclimatology, and macroecology, in order to better explore the historical changes of floristic realms and the underlying drivers.

## Comparison with Takhtajan's floristic regionalization and the updated Wallace realms

Several notable differences are recognized between the Takhtajan (1986) and our floristic maps (Supplementary Fig. 11). First, the "Paleotropical kingdom" in Takhtajan's map is divided into the Indo-Malesian realm and the African realm. These two realms have been separated by the Indian Ocean since the late Jurassic[31,37], which may have led to the division between them. The temporal changes in the floristic similarity between these two realms are also supported by woody angiosperm fossils in these regions[36].

Second, the "Antarctic kingdom" in Takhtajan's map is divided into the Chile-Patagonian and Novozealandic realms here, which is consistent with the view of Cox[1]. Our results indicate that the floras in these two regions are phylogenetically more similar to their adjacent realms than to each other throughout the geological history (Fig. 1b), which is also supported by the similarity in woody fossils during the Cenozoic[36].

Third, we define the new Saharo-Arabian realm, which was treated as a subset of the "Holarctic kingdom" in Takhtajan's map[4]. Our updated Holarctic realm mainly covers the ancient Laurasia landmasses[19,31]. The newly defined Saharo-Arabian realm covers northern Africa and the Arabian Peninsula, and has been connected with Africa and separated from the ancient Laurasia landmasses by the Tethys Sea since the Cretaceous[37,45]. Our analysis on realm dynamics shows that the flora in the Saharo-Arabian realm already differed from that in the Holarctic realm during the late Cretaceous (Fig. 2). Since the Early Miocene (23–16 Ma), the Saharo-Arabian realm experienced several waves of aridification, which may have further led to the evolutionary divergence of its flora from that of the Holarctic realm[46,47]. Notably, the separation of the Saharo-Arabian realm from the Holarctic realm is robust to different assumptions on the crown age of angiosperms (Supplementary Fig. 3), variations in phylogenetic topology (Supplementary Figs. 4–8), sampling biases (Supplementary Fig. 9), taxonomic beta diversity (Supplementary Fig. 10) and the chosen of different clustering methods (Supplementary Fig. 17). Even though, the boundary between the Saharo-Arabian and the Holarctic realms remains uncertain, as the fuzzy c-means clustering suggests that the boundary might encroach into Europe (Supplementary Fig. 17). The

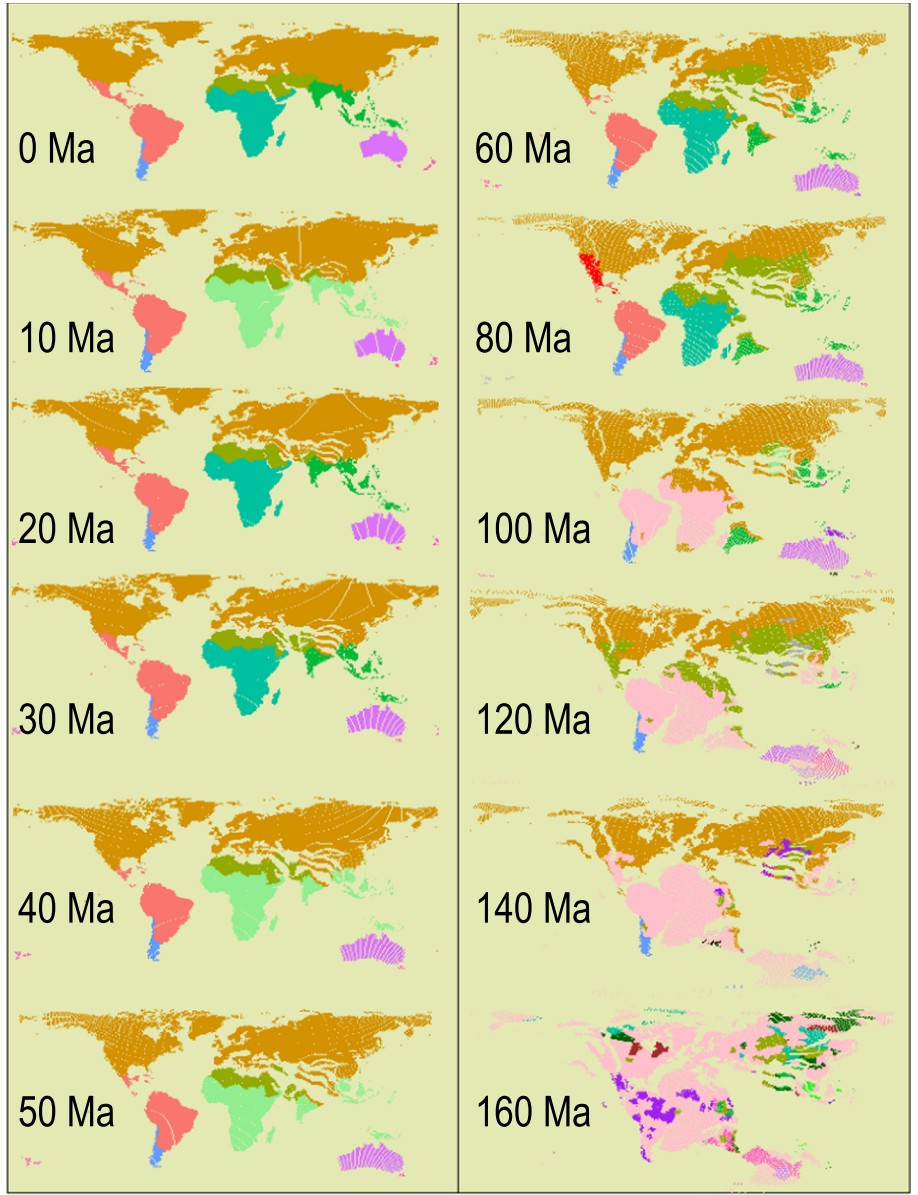

**Fig. 2 | Chronology of the present-day floristic realms.** The phylogenetic tree is cut at successive phylogenetic depths and all descendent leaves are collapsed into the branches encountered at that depth. Then the realms at each phylogenetic depth are identified using the same clustering method as in Fig. 1. It is noteworthy that we do not intend to estimate the ancestral geographic ranges of phylogenetic branches. This chronological sequence of maps represents the divergence times of flora that survived to the present day, but it provides only limited information on the ancestral floristic relatedness, which should be evaluated by fossils. The floristic realms which can be matched to the present-day realms are shown in the same colors as shown in Fig. 1a. As the present-day floristic realms are not distinguishable in some historical periods, we used other colors to represent these ancestral floristic realms. Specifically, light green in maps of 10, 40, and 50 Ma represents the ancestral realm covering the geographic ranges of the present-day African and Indo-Malesian realms; pink appearing from 100 Ma to 140 Ma represents the ancestral realms covering the present-day Neotropical+African realms, the present-day Neotropical+African+Indo-Malesian realms, and the present-day Gondwanan super-realm, respectively. Notably, most present-day floristic realms are undistinguishable in 160 Ma.

boundary uncertainty may be induced by the overlap between the Saharo-Arabian and Holarctic floras (Supplementary Data 5, also see Supplementary Fig. 19), and therefore, future investigations at regional scales are needed to further clarify the northern boundary of the Saharo-Arabian realm.

Fourth, the Cape region is ranked as one of six "kingdoms" in Takhtajan's map[4], but as a sub-realm of the African realm here. Our result is consistent with Cox[1]. The Cape region has not been geographically separated from Africa and shares similar tectonic history with the African continent[31]. The endemism in the Cape flora is primarily observed at species level, while the number of endemic taxa at higher levels, e.g., genus and family, is much lower than in other realms[1,48]. Many genera with high proportions of endemic species in the Cape region are also widely distributed in Africa, such as Erica, Protea, Helichrysum[49]. However, most endemic genera in the Cape region contain only very few species[1]. These findings suggest that the Cape flora may not have higher evolutionary distinctiveness at higher taxonomic (e.g., the genus or family) levels compared with floras in other African regions[1].

A comparison between our floristic map and the recently updated zoogeographic map (9) indicates several consistencies, suggesting that there are common drivers of terrestrial plant and animal biogeographic patterns. Specifically, the Saharo-Arabian and Indo-Malesian realms are identified in both maps and the boundaries of these realms

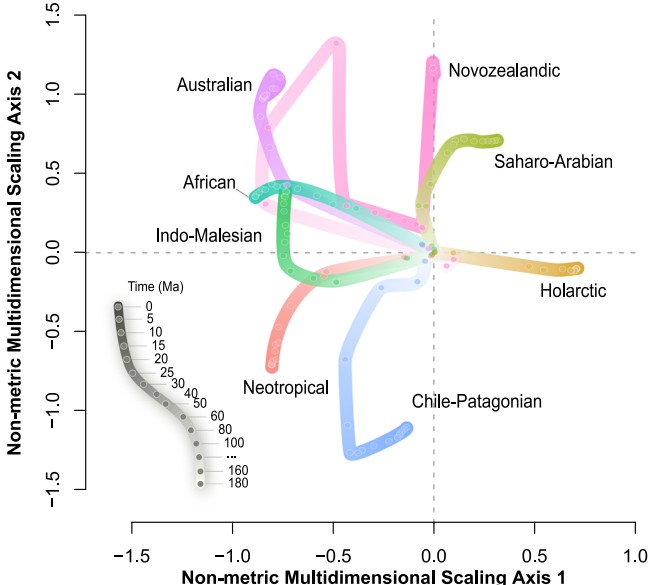

**Fig. 3 | The sequential appearance of the present floristic dissimilarities between different realms evaluated using non-metric multidimensional scaling (NMDS) ordination.** The Euclidean distance between dots is positively associated with the dissimilarity in phylogenetic composition of the flora between them: the larger the distance, the higher is the dissimilarity. Each line represents the distance of present phylogenetic composition of a realm to other realms at different phylogenetic depths. The colors of the lines are consistent with the colors of the realms shown in Fig. 1a, and the color gradient of each thick line represents evolutionary time. Source data are provided as a Source Data file.

are largely consistent (Supplementary Fig. 11). Moreover, the Cape region is not recognized as a realm in both maps. Notably, there are also interesting differences between our floristic and the zoogeographic maps. Our floristic map supports: (1) a Holarctic realm rather than the separated Palearctic, Nearctic and Sino-Japanese realms in the zoogeographic map; (2) an African realm rather than separated Afrotropical and Madagascan realms; (3) an Indo-Malesian realm rather than separated Oriental and Oceanian realms. These differences may to some extent reflect the effects of different dispersal abilities of plants and vertebrates on biotic interchanges between regions[1]. Fossil and molecular evidence suggests that dispersal across land bridges separated by water is relatively common and has occurred in many plant clades[39,40]. In contrast, long-distance dispersal across sea water in vertebrates has been found to be biased to clades with specific locomotion, e.g., flight in birds[50].

### The drivers on the division between floristic realms

Contemporary climate explains considerable variations in phylogenetic beta diversity within the Neotropical ($R^2 = 17.8\%$), Chile-Patagonian ($R^2 = 32.9\%$), Indo-Malesian ($R^2 = 23.8\%$), African ($R^2 = 14.7\%$) and Australian realms ($R^2 = 17.6\%$), but not within the Holarctic ($R^2 = 6.6\%$) and Saharo-Arabian realms ($R^2 = 1.3\%$). In contrast, contemporary climate has consistently extremely low explanatory power on the phylogenetic beta diversity between realms ($R^2 < 8\%$ for all realm pairs). These results suggest that, although contemporary climate influences floristic variations within some realms, it is not a dominant driver for realm division (Fig. 4).

We then evaluated the relative effects of the other two factors on the division of realms, i.e., the historical climatic differences across space during geological times (historical climate hereafter) and historical geographic isolation induced by plate tectonics (geographic isolation hereafter). It is noteworthy that geographic isolation and historical climate may, to some extent, interlink with each other as plate tectonics may lead to shifts in landmasses and their climates. Our results indicate

that the correlations between temporal dynamics in historical climate and geographic isolation are generally low in most cases (Supplementary Fig. 12, Pearson $r < 0.3$) except in the comparison between the Australian and Neovozealandic realms (Supplementary Fig. 12f, Pearson $r = 0.56 \pm 0.01$). To better compare the effects of historical climate and geographic isolation on realm division, we conducted hierarchical partitioning analysis to estimate their independent effects (Fig. 5). We find that historical climate has weak effects on the division between most realms. For the division of the temperate realms (i.e., Saharo-Arabian, Chile-Patagonia, and Novozealandic realms, Fig. 5b, d, f), the effect of historical climate increased from the Oligocene to the present. This may be because the global climate started to become cooler and drier since the late Eocene and this trend intensified after the mid-Miocene[31]. Compared with historical climate, geographic isolation has stronger effects on the division between the Gondwanan and Laurasian super-realms (Fig. 5a), between the Saharo-Arabian and Holarctic realms (Fig. 5b), between the (Neotropical, Chile-Patagonian) and the (African, Indo-Malesian) realms (Fig. 5e), and between the (Australian, Novozealandic) realms and other realms of the Gondwanan super-realm (Fig. 5c). These results suggest that geographic isolation induced by plate tectonics has played a dominant role in the division of these super-realms and realms.

Geological evidence indicates that the ancient Tethys Seaway separated the Gondwana and Laurasia landmasses before the Cenozoic[31,37], which may have led to the dominant effects of geographic isolation on the division between the Laurasian and Gondwanan super-realms (Fig. 5a) and on the division between the Holarctic and Saharo-Arabian realms (Fig. 5b). The breakup of Gondwana and the opening of the Atlantic Ocean may have enhanced the effects of geographic isolation on the division between the realms within the Gondwanan super-realm. (Fig. 5c, e, f). The present-day floras in South America and Africa cannot be distinguished from each other in the Middle Cretaceous (100 Ma, Fig. 2), but increasingly diverged from each other since 80 Ma (Fig. 3), which is possibly due to the reduced flora exchange caused by the expansion of the Atlantic Ocean as shown by the fossil evidence[29,31]. Hence the effect of geographic isolation induced by the expansion of the Atlantic Ocean on the division of the Neotropical and the Chile-Patagonian realms from African and Indo-Malesian realms within the Gondwanan super-realm also increased over time (Fig. 5e). The northward drift of the Australian plate and the southward drift of the Antarctic plate starting from the late Cretaceous combined with the onset of the Antarctic glaciation in the Oligocene (30–28 Ma) cut off the floristic exchange between Australia and other Gondwana landmasses (i.e., South America and Africa)[31]. This may have led to an increased effect of geographic isolation from the Oligocene to the mid-Miocene on the separation of the Australian and Novozealandic from the African and Indo-Malesian realms (Fig. 5c). New Zealand drifted away from the ancient Gondwana in the Late Cretaceous (80 Ma)[29], likely leading to a higher contribution of geographic isolation on the evolution of its flora than historical climate before the Eocene (Fig. 5f).

Geographic isolation has weaker effects on the division between the Neotropical and the Chile-Patagonian realms than historical climate through geological times (Fig. 5d). These two realms have been geographically connected during most of the Cenozoic period, which may have led to the weak effects of geographic isolation. Interestingly, neither geographic isolation nor historical climate well explain the division between the African and Indo-Malesian realms (Fig. 5g). This may be due to biotic interchange between the floras of African and the Indo-Malesian realms. The northward drift of the Indian subcontinent since the early Cenozoic and its final collision with Eurasia brought a large number of floristic elements that are closely related to the African flora to the Indo-Malesian realm[37,38], which may have led to the low explanatory power of both geographic isolation and historical climate on the division between these two realms.

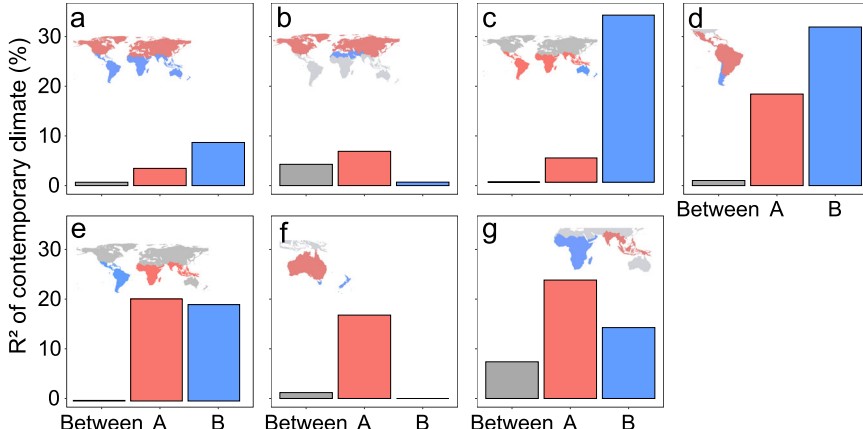

**Fig. 4 | Effects of contemporary climate on the phylogenetic beta diversity within and between floristic realms (or clusters of realms).** The bars show the explained variance in the phylogenetic beta diversity within (red and blue) and between (gray) different realms (or clusters of realms) as shown by the inset maps. **a** Gondwanan super-realm vs. Laurasian super-realm; **b** Holarctic realm vs. Saharo-Arabian realm; **c** (Australian, Novozealandic) realms vs. ((African, Indo-Malesian), (Neotropical, Chile-Patagonian)) realms; **d** Neotropical realm vs. Chile-Patagonian realm; **e** (African, Indo-Malesian) realms vs. (Neotropical, Chile-Patagonian) realms; **f** Australian realm vs. Novozealandic realm and **g** African realm vs. Indo-Malesian realm. Source data are provided as a Source Data file.

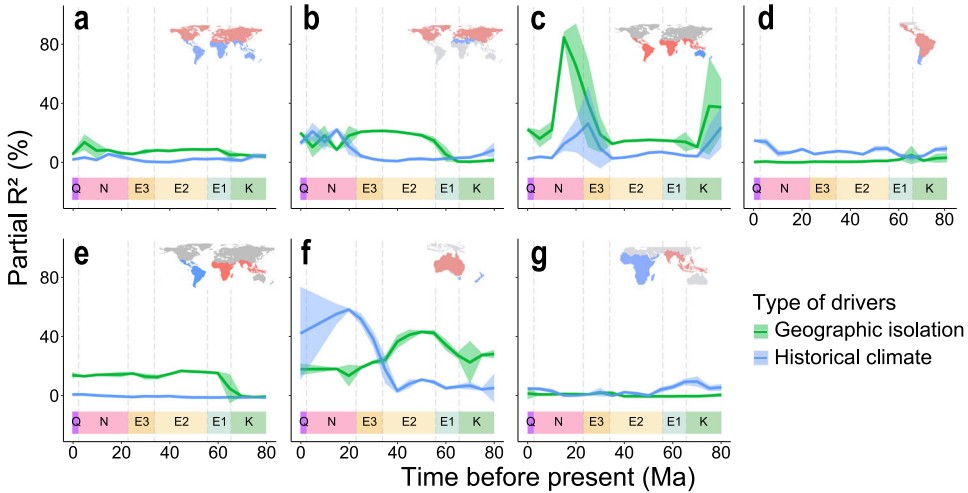

**Fig. 5 | Temporal changes in the relative effects of geographic isolation and historical climate on the phylogenetic beta diversity between floristic realms (or clusters of realms).** At each time interval of 1 Ma during the last 80 Ma, the partial $R^2$ of geographic isolation and historical climate are evaluated using hierarchical partitioning model as ln (phylogenetic beta diversity) ~ ln (Climate isolation) + ln (Geographic isolation). Colors of lines represent the independent $R^2$ of geographic isolation (green) and historical climate (blue) on phylogenetic beta diversity, respectively. The lines and the shaded areas represent the mean ± SE of the $R^2$ summarized every 5 Ma. **a** Gondwanan super-realm vs. Laurasian super-realm; **b** Holarctic realm vs. Saharo-Arabian realm; **c** (Australian, Novozealandic) realms vs. ((African, Indo-Malesian), (Neotropical, Chile-Patagonian)) realms; **d** Neotropical realm vs. Chile-Patagonian realm; **e** (African, Indo-Malesian) realms vs. (Neotropical, Chile-Patagonian) realms; **f** Australian realm vs. Novozealandic realm and **g** African realm vs. Indo-Malesian realm. Source data are provided as a Source Data file.

## The effects of clade evolution on the division of floristic realms

The clades with high contribution to realm divisions vary among realms. Specifically, we find that younger clades generally have higher contribution than older ones to the division of the (Australian, Novozealandic) realms from the other realms of the Gondwana super-realm (Supplementary Fig. 13c), the division between the (Neotropical, Chile-Patagonian) and the (African, Indo-Malesian) realms (Supplementary Fig. 13e), and the division between the African and Indo-Malesian realms (Supplementary Fig. 13g). This result corroborates the enhanced effect of geographic isolation caused by the breakup of Gondwana on realm divisions (Fig. 5). Among major angiosperm clades, malvids, campanulids and lamiids have high contributions to the divisions between African and Indo-Malesian realms ($R^2 > 89.8\%$), between Neotropical and Chile-Patagonian realms ($R^2 > 38.0\%$) and between Holarctic and Saharo-Arabian realms ($R^2 > 54.7\%$), respectively (see Supplementary Data 3 for details). Notably, our study only

indicates the contribution of present-day clades to global realm divisions. Further investigations could compare the relative contribution of extinct and present-day clades by integrating contemporary and fossil evidence.

We present the global map of floristic regionalization quantitatively delineated using the distributions and phylogeny of global angiosperm genera. Global lands are divided into two super-realms and eight realms. The boundaries and the hierarchical divisions between the identified realms mostly reflect the effect of geographic isolation induced by plate tectonics over geological times rather than the effect of contemporary and historical climate. These findings together with the high consistency between the boundaries in floristic and zoogeographic realms suggest that geographic isolation during the geological history is likely a common driver for the formation of both floristic and zoological realms. Our global map of floristic realms provides a geographic framework for a wide variety of comparative

studies in historical and ecological biogeography, macroecology, and systematics.

## Methods

A full description of the methods, such as phylogenetic reconstruction, compilation of distribution data, identification of floristic realms and sub-realms, and evaluation on the sensitivity of realm boundaries, is included in the Supporting Methods.

### The phylogeny of global angiosperm genera

A genus-level phylogeny for seed plants was constructed using molecular data for 8 gene markers obtained from GenBank (May 19, 2018), including 18S rDNA, ITS (i.e., ITS1, 5.8S ribosomal DNA and ITS2), and 26S rDNA from the nuclear genome; atpB, matK, ndhF and rbcL from the chloroplast genome; and matR from the mitochondrial genome. Sequences from hybrids and taxa with dubious identification were excluded. To construct the genus-level dataset, we first assessed the monophyly of each genus following[51]. For monophyletic genera, one representative sequence per marker per genus (generally the longest one) was selected. For a non-monophyletic genus (totally 593 genera, 4.7%), we only selected species from its core or the largest monophyletic clade. This procedure ensured that we only combined sequences from species belonging to the same monophyletic lineage. Accession number of the sequences that were used in our molecular analyses are available in Supplementary Data 6.

The genus-level sequences were aligned within each order separately and then merged using MAFFT v7.4 with the most accurate L-INS-i strategy[52]. Phylogenetic analysis were partitioned by RAxML v8.0.26[53] with GTRGAMMA model. We constrained the phylogenetic analyses in RAxML v8.0.26 using the APG IV relationships among angiosperm orders and among eudicots, monocots and magnoliids. The tree was dated with treePL v1.0[54] using fossil calibrations from ref. [55]. As the crown age of angiosperms is still debated[56], we conducted three dating analyses with different constraints on the age of angiosperm crown: (1) between 149 Ma and 256 Ma following[57]; (2) between 140 Ma and 210 Ma following[58]; and (3) between 140 and 150 Ma following[59]. Seed plant genera without sequence data were added to the dated phylogenies as polytomies based on current taxonomy, and then were resolved using the polytomy resolver following[60]. The final molecular and full phylogenies contain 12,539 and 14,244 seed plant genera, respectively. As all results are consistent across the phylogenies, we reported results based on the phylogeny with a constraint of 140–210 Ma for angiosperm crown age, and others in the supplementary materials.

To further explore the potential influence of the fast-evolving genes (particularly ITS1 and ITS2) on phylogenetic topology, we reconstructed the molecular phylogeny using sequence data without ITS and dated it in the same way as previously described. Then we compared it with the phylogeny based on the full sequence dataset and found both the topologies and phylogenetic distances among genera to be highly consistent between the two phylogenies (see supplementary method for details).

### Global distributions of angiosperm genera

Geographic distributions of angiosperm species were compiled from >1100 sources, including published regional and local floras, floristic investigations, specimen records and online databases (see Supplementary Data 1 for the full list of data sources). The geographic standard units (hereafter GSUs) used for the compilation of species distributions were generated following[61], and the average size of GSUs was ca. 4° longitude × 4° latitude. After removing small islands (<25,000 km²) and Antarctica, the earth's landmass was divided into 420 GSUs (see Supplementary Data 2). We classified the raw distributional data into four types: coordinates, range maps, gridded

distributions, and recorded localities. Depending on the types of the raw data, we applied different methods to reduce spatial conflicts between the original records and the boundaries of the GSUs used in our study and to improve the accuracy of species distributions in the final dataset (see Supplementary Methods for details of these methods). To improve the quality of species distribution data, we set a threshold for the number of data sources to keep an occurrence record of a species in a given GSU. For geographical units in Europe, Australia, China, South Africa, Madagascar and North America, an occurrence record of species in a geographical unit corroborated by at least 3 data sources was retained, leading to high confidence of the data quality in these regions. For the geographical units in Central America, Greenland, Amazon and Turkey, an occurrence record of species in a geographical unit corroborated by at least 2 data sources was retained, leading to medium confidence of the data quality in these regions. The entire data was retained for India, North and Central Africa, and Patagonia because of data deficiency in these regions, leading to relatively low confidence of the data quality in these regions.

The taxonomic status and the accepted names of species from all data sources were standardized following the World Flora Online (WFO, http://www.worldfloraonline.org/, accessed: December, 2022), Catalog of Life (COL, https://www.catalogueoflife.org/, accessed: May, 2018), ThePlantList (TPL, http://www.theplantlist.org/, accessed: Jan 3, 2015) and POWO (accessed: December, 2022). Synonyms are replaced with the accepted names. We kept accepted names with the highest confidence level. Taxonomic names that were identified as 'unresolved' in both COL and POWO were removed. The misspelt taxonomic names were corrected using the Taxonomic Name Resolution Service 4.0 (TNRS, https://tnrs.biendata.org/), which has been widely used in plant studies.

During data compilation, we also collected the status of species (i.e., being native, cultivated, introduced, invasive and hybrid) from regional data sources as much as we could, and non-native and hybrid species in different regions with clear records were not included in the database. After the compilation of distribution data at species level, we further checked the distribution maps and removed cultivated records from the database following the Plants of the World Online (POWO, https://powo.science.kew.org/, accessed: May, 2019) and efloras (http://www.efloras.org/, accessed: May, 2019).

Finally, we compiled the distributional data for each genus by aggregating distribution data of all its species. The distribution maps of all genera were carefully verified manually to improve data quality. We then integrated the phylogeny and distribution data, and the final distributional database contains 384,771 records for 12,664 angiosperm genera (see Supplementary Data 2), representing 90.63% of the total 13,974 accepted genera in POWO[62].

### Taxonomic and phylogenetic beta diversity

Simpson beta diversity[63] was used to evaluate the dissimilarity between species assemblages of GSUs:

$$\text{Beta diversity} = 1 - \frac{a}{a + \min(b,c)} \qquad (1)$$

where $a$ represents species shared between two GSUs, $b$ and $c$ represent species unique to each GSU. We used this metric because it is not affected by the number of species and provides unbiased estimate of compositional turnover across space[2,13]. Using Equ. (1), pairwise matrices of taxonomic beta diversity (calculated using the number of shared ($a$) and unique ($b$, $c$) species between two GSUs) and phylogenetic beta diversity (calculated based on the length of shared ($a$) and unique ($b$, $c$) branches of angiosperm phylogenies between two GSUs) were generated separately.

## Identification of floristic realms and sensitivity analysis of realm boundaries

Hierarchical clustering analyses were conducted using the "hclust" function in stats (version 3.6.2) package in R 3.6.1[64] to group GSUs into floristic super-realms, realms, and sub-realms. With this method, GSUs with highest similarity (i.e., lowest distance) were first grouped together and then the most similar groups were grouped into clusters. This process was repeated until all GSUs were all grouped into a single cluster. The hierarchical clustering analyses were conducted for taxonomic and phylogenetic beta diversity between the GSUs, separately.

Following Holt et al.,[2] we compared the performance and accuracy of different clustering methods. Performance evaluation aims to choose the clustering algorithm that can best represent the floristic divergences with the lowest number of clusters. For each clustering method, we calculated the proportion of beta diversity explained by identified clusters at a certain dendrogram height ($P_{beta}$) as the sum of between-cluster beta diversity divided by the sum of beta diversity between all GSUs[2]. The best performing method was identified as the one returning the minimum number of clusters when $P_{beta}$ reached 99%. Accuracy evaluation aims to choose the clustering algorithm that can represent the floristic divergences with the lowest biases. To do this, the co-phenetic correlation coefficients were calculated for each clustering algorithm using the "cophenetic" function[65] in stats (version 3.6.2) package in R 3.6.1[64], and the clustering method with the highest accuracy has the highest co-phenetic coefficients.

We found that the "average" method, also known as the Unweighted Pair Group Means Algorithm (UPGMA) performed the best (Supplementary Fig. 14). Using the UPGMA, an unrooted dendrogram was generated. Floristic realms and sub-realms were identified by cutting the dendrogram at different heights, which were determined following the approach of[2]. Floristic realms and sub-realms were identified as the clusters required to reach $P_{beta}$ = 80% and $P_{beta}$ = 95%, respectively[2]. Then we used the Non-metric Multidimensional Scaling (NMDS hereafter) to illustrate the relationships between floristic realms in a two-dimension non-hierarchical space.

The GSUs on realm boundaries may contain mixed floristic components, which may lead to soft boundaries[19,31]. For comparison with the hierarchical clustering analysis, we redefined the floristic realms using the fuzzy c-means clustering method. Unlike the hierarchical method, which assigns GSUs exclusively to one cluster, the fuzzy c-means clustering method estimates the likelihood of each GSU belonging to a certain cluster, under a given degree of fuzziness[66]. We then conducted silhouette analysis for both results of UPGMA and fuzzy c-means clustering to evaluate the uncertainties in assigning GSUs to a single floristic realm[67]. The identification of floristic realms may also be influenced by uncertainties in phylogenetic topology and incomplete sampling across lineages[24]. Therefore, we repeated the hierarchical clustering analysis using: (1) the single maximum credibility tree; (2) the trees containing only genera with molecular data, (3) randomly sampled post burn-in posterior trees from the polytomy resolver, and (4) trees containing only the genera used in a recent study[24].

## Changes in floristic realms through time

To explore the phylogenetic depth at which the spatial divergence of realms appears, we cut the phylogenetic trees at different geological times and generated the maps of floristic realms at each time following the approach used in refs. 34, 68–71. This approach collapsed all the descendent leaves of each branch encountered at a given time and then identified the floristic realms using the UPGMA clustering method as described above. We then illustrated the changes in the dissimilarity between the floras of different realms by overlaying the NMDS ordinations conducted at different geological times. Note that this analysis did not intend to estimate the ancestral geographic ranges of phylogenetic branches or the ancestral floristic assemblages.

## Effects of contemporary climate, historical climate, and geographic isolation on realms divisions

If the contemporary climate dominates the division between floristic realms, contemporary climate should have higher explanatory power on the beta diversity between than within realms. The climate difference of a given pair of GSUs was defined as the Euclidean distance between them in a two-dimensional (i.e., annual mean temperature and precipitation) climatic space. Then we evaluated the effect of contemporary climate on beta diversity between and within realms using Ordinary Least Square (OLS) regressions with phylogenetic beta diversity as the response variable and contemporary climate differences as the predictor.

Following the method in ref. 72 and the reconstructed map in ref. 73, we produced paleogeographic maps with a spatial resolution of 1 × 1 degree from 80 Ma up to the present in a 1 Ma step. Then, at every geological time, we calculated the geographic isolation between each two grid cells as the minimum total route cost between them, which was a function of geographic distance, separation by ocean, and elevation of the grid cells on the route. The geographic isolation was estimated using the 'gdistance' R package (version 1.6)[74]. Paleo-temperature and paleo-precipitation at 1 Ma step from 80 Ma to the present were reconstructed by ref. 72. We calculated the Pearson correlation of geographic isolation and historical climate differences for each pair of floristic realms from 80 Ma to the present, and then hierarchical partitioning analysis was used to compare the relative effects of geographic isolation and historical climate at each geological time on the division of floristic realms. Specifically, phylogenetic beta diversity between GSUs of two realms was used as the response variable, and geographic isolation and historical climate distances between GSUs were used as the predictors.

## Contribution of different clades to floristic realms division

We first detected angiosperm clades whose descendant lineages have little overlap in their geographic distributions, and then measured their contribution to the division of floristic realms as the degree of consistency between the realm boundaries and the geographic divergences of their descendant lineages. Specifically, the geographic divergences of descendant lineages of a clade was evaluated using node-based analysis conducted in the "nodiv" R package (version 1.4.0)[34]. Specifically, this method calculates the degree of mismatch in the geographic distribution of two sister lineages diverging at a given node, i.e., the 'geographic node divergence' (GND) score, and the specific overrepresentation scores (SOS) for each geographic unit occupied by the two sister lineages. GND values over 0.65 indicate significant distributional divergence between the two sister lineages. Positive (or negative) SOS values indicate predominance of one of the two descendant lineages in a geographic unit (see Supplementary Fig. 15). Using node-based analysis, we detected the clades with GND scores over 0.65 and extracted the SOS scores of all GSUs occupied by these clades. The contribution of each of these clades on the division of two floristic realms was measured by the $R^2$ of the one-way analysis of variance (ANOVA) with SOS scores as the response variable, and the two realms as the predictor.

## Reporting summary

Further information on research design is available in the Nature Portfolio Reporting Summary linked to this article.

# Data availability

All data needed to evaluate the conclusions in the paper are present in the paper and/or the Supplementary Materials. The distribution data can be found in Supplementary Data 2. The phylogeny and the

shapefiles of the floristic realms identified in this paper are available at https://en.geodata.pku.edu.cn/index.php?c=content&a=list&catid=199 (User: flowertree; Password: flowertree). DNA sequence data were downloaded using from GenBank (as of May 19, 2018) and the Accession numbers for DNA sequences can be found in Supplementary Data 6. Global Administrative Areas boundaries were downloaded from http://www.gadm.org (accessed: May 2016). Distribution data was obtained from both on-line databases and directly from the literature and the complete list of distributional data sources is provided as supplementary data. Species distribution data recorded as locality names were searched in the global geographical names service http://www.geonames.org. The taxonomic status and the accepted names of species from all data sources were standardized following the *World Flora Online* (WFO, http://www.worldfloraonline.org/, accessed: December, 2022), *Catalog of Life* (COL, https://www.catalogueoflife.org/, accessed: May, 2018), *ThePlantList* (TPL) available at http://www.theplantlist.org/ (accessed: Jan 3, 2015) and POWO (accessed: December, 2022). Climate data was downloaded from the WorldClim database v2.0 (https://www.worldclim.org/, accessed: December, 2022). Paleo-digital elevation models are obtained from Scotese's paleoatlas[73]. Paleoclimate data are obtained from[72]. Source data are provided with this paper.

## Code availability

All code needed to evaluate the conclusions in the paper can be found in https://github.com/yunpengliu1994/regionalization (https://doi.org/10.5281/zenodo.7758185)[75].

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

## Acknowledgements
This work was supported by the National Key Research Development Program of China (#2022YFF0802300), National Natural Science Foundation of China (#31988102, #32125026, #31770566), and the Strategic Priority Research Program of Chinese Academy of Sciences (#XDB31000000). D.D received additional support by the Norwegian Metacenter for Computational Science (NOTUR; project NN9601K). MKB, CR and YL acknowledge the Danish National Research Foundation (DNRF96) and VILLUM FONDEN (25925) for support of the Center for Macroecology, Evolution and Climate.

## Author contributions
Z.W., C.R., and Y.L. conceived the idea; Z.W. Y.L., X.X., X.S., and A.L. constructed the distribution dataset; Z.W., X.X., and D.D. conducted the phylogeny; L.P. and N.Z. conducted the palaeogeographical and paleoclimate data; Y.L. and M.B. conducted the node-based analysis; Y.L. and N.S. generated the figures in the manuscript; Y.L. led the analysis and writing, and all authors contributed to the writing.

## Competing interests
The authors declare no competing interests.
