## [Peer Review File · Nature Communications]

An updated floristic map of the worldREVIEWER COMMENTS

Reviewer #1 (Remarks to the Author):

Review of "The updated floristic map of the world"

The authors present an updated floristic map of the world based on hierarchical clustering of 85% of all known angiosperm genera using phylogenetic beta-diversity metric. Their work is global at scale which is influential and the product presented in the paper has a big impact to large scale biodiversity studies where the units the authors provide could be used in future studies as study units in science, conservation, global change, etc. The work is a significant contribution to the field (and related fields), and is a step forward from previous studies. The conclusions presented are supported by the data. Some questions are raised in my review that I would like expanded on/revised on. The work is of good quality and enough detail is provided.

The introduction goes through the origin of biogeographic regionalisation (early works and influencers) and then describes previous work on floristic schemes by previous authors. The most recent global work on plants was by Carta et al. (2022) cited and referred to in detail in the paper who analysed phylogenetic beta-diversity at species level with 20% level sampling and produced a map of three floristic kingdoms. The analyses presented here provides a large increase in sampling based on generic level sampling of 85% of all angiosperm genera which enables a different depth of view. Results are compared to previous floristic works and the zoogeographical regionalisation by Holt et al. (ref 2) which was an analyses based on phylogenetic relatedness of birds, mammals and amphibians. Authors then explore the effect of climate on the boundaries of the floristic regions they find and explore the effect of particular clades/lineages on these boundaries to see if some are driven by particular families/order/clades only.

The analyses through time of the floristic realms is intriguing, and I appreciate that the authors provide clear caveats of the method in Fig. 2 legend relating to the lack of ancestral range estimation. I think presenting these results still has value, as the caveats have been clearly stated. Figure 1 could be enhanced by using colour from the map for each region. I really like Figures 2-4, they are informative.

I checked the underlying phylogeny used in the analysis for Solanaceae (family I know well) and the generic level relationships in the tree are realistic representations of relationships based on current expert validated data as far as I can see which gives confidence. What I would like to see detail and justification for is the inclusion of fast evolving markers such as ITS for building a phylogenetic tree across all angiosperms: alignment of ITS within a family can be troublesome, let alone across several order or all angiosperms. Local alignments per family could be used in my view but justifying a single angiosperm wide alignment of ITS is difficult if not impossible. What effect would removal of ITS have on the phylogeny? The phylogeny built in this study is sold as something big in the method so I would like to see more detail on the phylogeny whether families in APGIV were recovered monophyletic, and even potential revision of the phylogeny without ITS. The methods specify that distribution data for each genus was calculated by "aggregating distribution data of all its species." This would be complicated for large genera and genera that include cultivated/invasive species. Were the most commonly cultivated species excluded at this stage?

Fig. S11 legend should be modified to clarify that "Colours correspond to the floristic regions from this study as illustrated in Fig. 1, ..."

Lines 198-200: sentence refers to the same figure twice but different subfigures – please remove repetition and clarify which ones are needed.

Paragraphs on page 8 lines 197-211 discussing the effect of geography versus climate should acknowledge that the movement of continents (i.e. geography) also resulted in changes in climatic conditions on those continents so that these two factors are somewhat interlinked.

Because the spatial resolution of the study is country level (or major areas within large countries such as Brazil i.e. geographic standard units), the boundaries of these floristic units cannot really be critically evaluated in my view. For example Fig. S1 is somewhat meaningless in my view, because what does it mean that the boundaries of the Holarctic region show some uncertainty? That the membership of a country inside/outside of this region is not certain i.e. is ambiguous?

Reviewer #2 (Remarks to the Author):

I reviewed a previous version of this paper submitted elsewhere. I reiterate here my generally positive impression of this work, noting in particular the efforts the authors have invested in generating robust phylogenetic and geographical data at a global scale. I believe this is a useful contribution worth publishing in a high profile journal, and that it adds to recently published work by Carta et al. (cited in the text) exploring similar patterns using a smaller set of plant taxa.

I have only a few comments on this new submission; however, there remains one sticky point that the authors need to address – the sensitivity of the defined realms to the hierarchical clustering approach. Such clustering methods provide an over-confident estimate of cluster membership, and I had previously suggested alternative fuzzy-clustering methods that allow cells to be a member of more than one cluster, embracing the uncertainty in the data. Application of this methodology did not return similar results. The authors need to address this incongruence, otherwise we cannot be confident that the biogeographic regionalisations they present are not simply a product of the statistical method they use. Why do fuzzy clustering and hierarchical clustering return different results and why should we objectively prefer the latter over the former?

Minor comments:

1. I suggest a small edit to the title: "An updated ...".

2. As I mentioned in my previous review, the authors need to take care in how they describe historical patterns. From the distribution of extant taxa we cannot make strong inference about how realms were related in the past (we are missing now extinct taxa that may have brought them closer together or separated them in phylogenetic space). The phrasing in this new submission is generally improved; however, there are still a few places where the authors slip, for example (lines 125-126), "The floristic dissimilarity between these two realms peaked during the early Cenozoic and then decreased before the Pliocene". This may be true, but an analysis of extant taxa cannot support this statement. What the authors can show is the phylogenetic depth at which extant taxa in these regions share proportionally more or less evolutionary history. A mindful edit of the text (which is, in general, very well written) is all that is required.

3. A point I also raised earlier is on the information provided on the contribution of different clades. This is described in the methods (section awkwardly titled: Contribution of different clades to floristic realms division), but is not integrated into the main text of the Results & Discussion. If this is an interesting and informative analysis, then the authors should expand upon it, otherwise this section could be dropped or retired to the supplement.

REPLIES TO REVIEWERS' COMMENTS

Reviewer #1 (Remarks to the Author):

Review of "The updated floristic map of the world"

The authors present an updated floristic map of the world based on hierarchical clustering of 85% of all known angiosperm genera using phylogenetic beta-diversity metric. Their work is global at scale which is influential and the product presented in the paper has a big impact to large scale biodiversity studies where the units the authors provide could be used in future studies as study units in science, conservation, global change, etc. The work is a significant contribution to the field (and related fields), and is a step forward from previous studies. The conclusions presented are supported by the data. Some questions are raised in my review that I would like expanded on/revised on. The work is of good quality and enough detail is provided.

RESPONSE: Thank you very much for your effort reviewing our manuscript and for your positive comments. We have revised the manuscript following your suggestions, which has led to a substantially improvement of the manuscript. We are sorry for the delay in sending the manuscript back since it took almost two months to construct the phylogeny without ITS and to calibrate its divergence times. We provided point-by-point responses to your comments as following.

The introduction goes through the origin of biogeographic regionalisation (early works and influencers) and then describes previous work on floristic schemes by previous authors. The most recent global work on plants was by Carta et al. (2022) cited and referred to in detail in the paper who analysed phylogenetic beta-diversity at species level with 20% level sampling and produced a map of three floristic kingdoms. The analyses presented here provides a large increase in sampling based on generic level sampling of 85% of all angiosperm genera which enables a different depth of view. Results are compared to previous floristic works and the zoogeographical regionalisation by Holt et al. (ref 2) which was an analyses based on phylogenetic relatedness of birds, mammals and amphibians. Authors then explore the effect of climate on the boundaries of the floristic regions they find and explore the effect of particular clades/lineages on these boundaries to see if some are driven by particular families/order/clades only.

The analyses through time of the floristic realms is intriguing, and I appreciate that the authors provide clear caveats of the method in Fig. 2 legend relating to the lack of ancestral range estimation. I think presenting these results still has value, as the caveats have been clearly stated.

Figure 1 could be enhanced by using colour from the map for each region. I really like Figures 2-4, they are informative.

RESPONSE: Thanks for the suggestion. We revised the colour of Fig. 1 as suggested. Please see the new Figure 1.

I checked the underlying phylogeny used in the analysis for Solanaceae (family I know well) and the generic level relationships in the tree are realistic representations of relationships based on current expert validated data as far as I can see which gives confidence. What I would like to see detail and justification for is the inclusion of fast evolving markers such as ITS for building a phylogenetic tree across all angiosperms: alignment of ITS within a family can be troublesome, let alone across several order or all angiosperms. Local alignments per family could be used in my view but justifying a single angiosperm wide alignment of ITS is difficult if not impossible. What effect would removal of ITS have on the phylogeny? The phylogeny built in this study is sold as something big in the method so I would like to see more detail on the phylogeny whether families in APGIV were recovered monophyletic, and even potential revision of the phylogeny without ITS.

RESPONSE: Thanks for the comments. Following your suggestions, we conducted the following revisions and analyses. We included these descriptions in the manuscript. Please see lines 320-325 in the maintext, lines 29 – 87 in the supplementary method and the newly added Fig. S21.

First, in our analysis, the ITS data included internal transcribed spacer 1 and 2 (ITS1, ITS2), and the 5.8S ribosomal DNA (5.8S). Among these three loci, 5.8S was a slow-evolving gene and very conservative across higher taxonomic levels such as orders, while ITS1 and ITS2 were fast evolving genes. Because alignment of the ITS1 and ITS2 between divergent groups is difficult and may lead to unwanted artifacts, we adopted an alignment strategy with the following steps for the alignment of ITS1 and 2. 1) The sequences of each plant order were placed in a separate matrix and were aligned using the L-INS-i strategy in MAFFT with the following commands: --

localpair -- maxiterate 1000 --adjustdirectionaccurately. 2) The order-level alignments were put together as a single fasta file. For each of these combined fasta files, subMSA table file with the information on which sequences correspond to individual order-level alignments was created using the makemergetable.rb script distributed with MAFFT. 3) The order-level alignments were then aligned to each other in MAFFT using the --localpair --merge commands that allow alignment of multiple sequence alignments. It is worth mentioning that the separate order-level alignments will, although indirectly, introduce to some extent a soft constraint on orders monophyly even if some orders were not explicitly constraint to be monophyletic. This approach is consistent with the topological constraints that we used for the deeper nodes on the backbone topology and is better than other alternatives as it decreases alignment errors and ensures higher consistency with currently accepted taxonomy.

Second, following your suggestion, we reconstructed the phylogeny using sequence data without ITS conducted in RAxML, and dated it in the same way as reported in the main text. Then we compared it with the original phylogeny (see the newly added Fig. S21 in the Supplementary materials). We found that only a small proportion of genera (1,176 genera, 9.4% of the total 12,456 of angiosperm genera with molecular data) were dropped when removing ITS (see Fig. S21a for the proportion of dropped genera in each family). The dropped genera were mainly from a few very large families (Fig. S21b, e.g. Asteraceae, 267 out of 1454 genera are dropped), while the number of genera in 349 (82.5%) out of 423 families remain unchanged when removing ITS. Most importantly, the removal of ITS did not affect the overall topology of phylogeny. Specifically, we found that the topologies of 404 families (95.5% of all 423 families) remained the same after removing ITS. Some small changes in the topology of a few genera were found in the very large families. Moreover, we found that after removing ITS, the branch lengths of genera and the phylogenetic distances between genera, which is directly related with the estimation on phylogenetic beta diversity, were highly consistent with those estimated from the original phylogeny (Fig. S21c, d; for branch lengths of genera, Pearson $r = 0.81$; for phylogenetic distances between genera, Mantel $r = 0.965$, $p = 0.01$). These results suggest that the removal of ITS did not dynamically change the phylogeny and the estimation of phylogenetic beta diversity used for the division of biogeographical regions.

Third, since we did not constrain families monophyly we did check whether families as defined in the APGIV were recovered monophyletic by our analyses. From the 423 families

present in our dataset with molecular data, most were recovered as monophyletic in our phylogeny. Only 19 families (4.5%) were found to be nonmonophyletic and most of these cases are due to the position of a few (often a single) genera, while core family groups remain monophyletic. Families that were not recovered monophyletic are also families which generic composition has been contentious (e.g., Capparaceae). Therefore, we think that, although our phylogeny is not designed to resolve contentious phylogenetic issues across Angiosperm lineages, it offers a good global overview of the Angiosperm phylogeny at the genus level.

The methods specify that distribution data for each genus was calculated by “aggregating distribution data of all its species.” This would be complicated for large genera and genera that include cultivated/invasive species. Were the most commonly cultivated species excluded at this stage?

RESPONSE: Yes, the cultivated, introduced, and invasive species in different regions were excluded from the distribution database. We included more descriptions about the treatment of cultivated species and records. Please see line 164 – 171 in the Supplementary method. First, during data compilation, we collected the information of the status of species (being native, cultivated and hybrid) from different regional data sources, and the cultivated and hybrid species in different regions were not included in the database. After the compilation of the distribution data at species level were finished, we manually checked the distribution maps and removed the cultivated records of each species from the database following the *Plants of the World Online* (<https://powo.science.kew.org/>) and *efloras* (<http://www.efloras.org/>). Second, after merging the species distributions within each genus, we manually screened and checked the distribution maps of all genera again to improve the quality of the distribution data.

Fig. S11 legend should be modified to clarify that “Colours correspond to the floristic regions from this study as illustrated in Fig. 1, ...”

RESPONSE: Done.

Lines 198-200: sentence refers to the same figure twice but different subfigures – please remove repetition and clarify which ones are needed.

RESPONSE: Done.

Paragraphs on page 8 lines 197-211 discussing the effect of geography versus climate should acknowledge that the movement of continents (i.e. geography) also resulted in changes in climatic conditions on those continents so that these two factors are somewhat interlinked.

RESPONSE: Thanks for your comments and we agree. We conducted the hierarchical partitioning analysis to compare the relative effects of geographic isolation and climate distances between geographical standard units on the division of floristic realms. The results are shown in Fig. 5 and reflect the independent effect of geographic isolation and climate distances. Based on your comments, we further calculated the Pearson correlation of geographic isolation and climate differences for each pair of floristic realms from 80 Ma to the present. In most cases, the correlation coefficients were very low. We added these results as fig. S12 and included more discussion about the potential linkage between the movement of continents and paleoclimate change (lines 209-217, 411 -412 in the main text).

Because the spatial resolution of the study is country level (or major areas within large countries such as Brazil i.e. geographic standard units), the boundaries of these floristic units cannot really be critically evaluated in my view. For example Fig. S1 is somewhat meaningless in my view, because what does it mean that the boundaries of the Holarctic region show some uncertainty? That the membership of a country inside/outside of this region is not certain i.e. is ambiguous?

RESPONSE: Sorry for the unclear statements. We revised the statement on the legend of Fig. S1 and the text in the method (lines 375-378 in the main text and lines 271-292 in the Supplementary Method). The geographic standard units (GSUs) near the edges of the identified realms may contain mixed floristic components, which will lead to uncertainties in their membership belonging to a floristic region. Therefore, recent studies suggest that regions along the edges of the identified floristic realms may form a “transitional zone”. In our study, we used the silhouette analysis to evaluate the degree of ambiguous in the membership of GSUs belonging to the identified realms. The results indicate that only a few GSUs near the edges of realms are found to have high uncertainties in their membership, while most GSUs are assigned to a realm with high confidence.

Reviewer #2 (Remarks to the Author):

I reviewed a previous version of this paper submitted elsewhere. I reiterate here my generally positive impression of this work, noting in particular the efforts the authors have invested in generating robust phylogenetic and geographical data at a global scale. I believe this is a useful contribution worth publishing in a high profile journal, and that it adds to recently published work by Carta et al. (cited in the text) exploring similar patterns using a smaller set of plant taxa.

RESPONSE: Thank you very much for your effort reviewing our manuscript again. We have revised the manuscript following your suggestions, including comparing clustering results with fuzzy c-means, modifying descriptions on historical floristic patterns, and adding the discussion about the contribution of different clades to the main texts. We provided point-by-point responses to your comments as following.

I have only a few comments on this new submission; however, there remains one sticky point that the authors need to address – the sensitivity of the defined realms to the hierarchical clustering approach. Such clustering methods provide an over-confident estimate of cluster membership, and I had previously suggested alternative fuzzy-clustering methods that allow cells to be a member of more than one cluster, embracing the uncertainty in the data. Application of this methodology did not return similar results. The authors need to address this incongruence, otherwise we cannot be confident that the biogeographic regionalisations they present are not simply a product of the statistical method they use. Why do fuzzy clustering and hierarchical clustering return different results and why should we objectively prefer the latter over the former?

RESPONSE: Thanks for the comments. Following your suggestion, we included the comparison on the results and performance of fuzzy c-means (fig. S1 and the newly added fig. S20). Please see the statements in the main text (lines 100 -101, 375 – 378) and see the Supplementary Method (lines 295 - 303), and the Supplementary Results and discussions (lines 532 - 562) for detailed discussion about the comparison between the results based on the UPGMA and the fuzzy c-means methods.

Minor comments:

1. I suggest a small edit to the title: “An updated ...”.

RESPONSE: Thanks. Sounds better than the current title and we made the revision as suggested.

2. As I mentioned in my previous review, the authors need to take care in how they describe historical patterns. From the distribution of extant taxa we cannot make strong inference about how realms were related in the past (we are missing now extinct taxa that may have brought them closer together or separated them in phylogenetic space). The phrasing in this new submission is generally improved; however, there are still a few places where the authors slip, for example (lines 125-126), “The floristic dissimilarity between these two realms peaked during the early Cenozoic and then decreased before the Pliocene”. This may be true, but an analysis of extant taxa cannot support this statement. What the authors can show is the phylogenetic depth at which extant taxa in these regions share proportionally more or less evolutionary history. A mindful edit of the text (which is, in general, very well written) is all that is required.

RESPONSE: Thanks for the reminding. We revised this part very carefully as suggested and rephrased the sentences that may lead to misunderstanding. See lines 118-151,235-238 in the main text.

3. A point I also raised earlier is on the information provided on the contribution of different clades. This is described in the methods (section awkwardly titled: Contribution of different clades to floristic realms division), but is not integrated into the main text of the Results & Discussion. If this is an interesting and informative analysis, then the authors should expand upon it, otherwise this section could be dropped or retired to the supplement.

RESPONSE: Thanks for your comment. We integrated the contribution of different clades into the main text as suggested. See lines 262-275.

REVIEWER COMMENTS

Reviewer #1 (Remarks to the Author):

Review of the re-submitted "An updated floristic map of the world"

The authors present an enhanced version of their previous submission that presents an updated floristic map of the world. The revisions are good, but I have a major concern flagged up by a further inspection of generic distribution patterns Fig. S9, as well as a few points/questions regarding some other points.

The major concern relates to the distribution of genera that were used as input for analysis. The distribution of genera is shown in Fig. S9, with natural ranges of 12,778 genera used coloured by the realm to which they belong. Based on that figure, every single genus found in Australia is endemic to the continent, which simply cannot be the case. Similarly, every genus found in South America does not occur in Africa, which simply is not true. Based on this figure, there are no pantropical/global genera in the input data, which is extremely strange and difficult to explain, considering that nearly 10% of global plant diversity is found amongst 22 large plant genera that include >1000 species each and are mostly global in their distribution. Scale is missing from Fig. S9, which is needed to understand the scale used in each subfigure. This needs to be revised so that the reader understands how many genera were not informative i.e. were found across all realms, and how many across different pairs of realms. Perhaps a ven-diagram would be a good way of showing this detail.

Another aspect of the quality control would be the synonymy used for genera. I found several old synonyms in the dataset, e.g. *Lycopersicon* is a genus now nested in *Solanum*, but in the dataset of the paper, it was kept separate with a wide distribution which reflects cultivated species (tomato) which the authors argue they had dealt with having added methods removing non-natives. Other synonyms include *Symbegonia* (is now *Begonia*), etc. Please run your data through World Flora Online that get accepted generic names standardized would be my strong suggestion, if your current pipeline doesn't pick these obvious established synonyms at generic level. Plants of the World Online currently lists 13,973 accepted plant genera, which contrasts starkly against the number of 14,289 genera given in SI File page 7 methods section. The raw data should be ran against synonymy prior to all analyses on the distribution data. These synonymous names should not appear in the phylogeny used either.

Figure 1 is great, but I think this would be best visualised as an unrooted dendrogram because you cannot argue to know the outgroup. The software generally roots on the most distant branch, but this assumption to make the most distant group an outgroup is not really needed in your work. An unrooted dendrogram would be a better way to visualise the similarity. Use FigTree software for example generally used for molecular phylogenetics.

Figure 2 would benefit from colour legend explaining the realms. What does pink refer to? You state that only 19 families were found to be non-monophyletic: it would be important to list these somewhere at least in supplementary materials to understand which ones and to see the reasons for this. For example, knowing if it is Leguminosae and Asteraceae (two large families) due to overall polyphyly would be important.

The revised methods state that the distribution data for each genus was checked manually. There were 12,778 genera in the final dataset, how was the manual check actually done? This seems a fantastic addition but is hard to understand how this was achieved. How many cases errors were spotted and corrected? (SI material page 6-7).

You say you made distinct ITS alignments for orders. I see this is an advancement from previous general alignment of ITS across all angiosperms, but better option would be to chop the alignments into groups based on % divergence cut off. This is something for future work, I understand you have tried to do best possible approach.

The methods say that sources of species occurrence data, and it would be nice to see summary on data bias in terms of geography/realm/continent, or illustration of density of occurrence points used geographically in order to understand data quality.

Minor points:

The terms floristic and geographic are used, but sometimes as "geographical" (e.g.,

zoogeographical): I don't know if there is a difference, but I would think better to use the term geographic (not geographical). This applies to biogeographic too (not biogeographical).

P. 3 line 38: remove word "regionalization", not needed in my view

p. 3 line 41: remove "of the different regions", not needed

p. 4 line 72: descendant (correct spelling)

p. 4 line 74: edit to "to the biogeographic regionalization due to ..."

p. 4 lines 75-76: edit to "and geographic distributions (34)."

Page 4 line 80: edit to "between regions at genus level."

p. 5 first subheading: simpler heading "World's floristic regionalization"

p. 5 line 108: add "due to climatic or geographic barriers"

p. 5 lines 109-115: add a note that these results "suggest recent exchange through dispersal of lineages between the two regions".

p. 8 lines 209-214: first sentence of the paragraph is difficult, perhaps edit to simplify. Get rid of It's and write out It is. Plate tectonics resulted in landmasses shifting in their climate but also climates kept changing globally over millions of years so clarify that both factors means that geographic isolation and climates may be linked.

p. 12 line 324: edit to "to be highly consistent"

Reviewer #2 (Remarks to the Author):

The authors have invested a great deal of effort into their analyses, and I compliment them for their work.

The one important concern I raised previously was on the sensitivity of the identified realms to the choice of clustering algorithm. The authors now include results using fuzzy c-means, and this is very useful. I do not completely agree with their justification for preferring UPGMA over the alternative, which seems to be based more on practical rather than statistical justifications (suppl lines 552-562), but this is a minor point. More importantly, the comparison of methods allows us to better evaluate confidences in the delineation of realms. For example, the separation of the Chile-Patagonian realm reported using phylo beta diversity (but not taxonomic diversity) is not recognised using fuzzy c-means, and thus appears to be particular to the clustering method used. This is an important observation, and should be clarified in the main text (perhaps at paragraph lines 109-115).

One key result, highlighted in the abstract, is the recognition of a "new" Saharo-Arabian realm. However, as we see in the silhouette analysis, this is also the realm that has a boundary that is less well defined, suggesting there might still be some uncertainty with respect to the distinctness of this floristic unit. The uncertainty in the delineation of this 'realm' is again demonstrated in the clustering by fuzzy c-means, which suggests that it might encroach into the European flora, I strongly recommend that the authors embrace this uncertainty in their description of this geographic unit (e.g. lines 165-173). We may well have evidence for a distinctive floristic region, but its bounds are still unclear.

The authors may find additional statements on the delineation of realms that are not fully supported by both methods, and I encourage them to make this clear to the reader in the main text, where appropriate. By recognising where there is uncertainty, I hope the paper will become stronger, and also highlight regions where floristic affinities are still unresolved.

Minor point, to avoid confusion, I suggest the authors refer to 'present-day floras' versus 'extant floras' in the section titled "The divergent times between the identified realms". This is because we are looking at the ancestral affinities of the floras (extant) in the present day, not the floras that may have been extant in these regions historically.

REPLIES TO REVIEWERS' COMMENTS

Reviewer #1 (Remarks to the Author):

Review of the re-submitted "An updated floristic map of the world"

The authors present an enhanced version of their previous submission that presents an updated floristic map of the world. The revisions are good, but I have a major concern flagged up by a further inspection of generic distribution patterns Fig. S9, as well as a few points/questions regarding some other points.

RESPONSE: Thank you very much for your efforts on reviewing our manuscript again. We have revised the manuscript following your suggestions. We apologize for the misinterpretation of our data quality of species distributions due to the unclear Fig. S9. We carefully checked and corrected the genera names following the most current treatment of plant names (i.e. World Flora Online as suggested, <http://worldfloraonline.org/>). We also included more details about the process of species distribution data compilation and examination, and demonstrated the quality of our species distribution data in different regions. Moreover, we modified the dendrogram as suggested. We provide point-by-point responses to your comments as following.

The major concern relates to the distribution of genera that were used as input for analysis. The distribution of genera is shown in Fig. S9, with natural ranges of 12,778 genera used coloured by the realm to which they belong. Based on that figure, every single genus found in Australia is endemic to the continent, which simply cannot be the case. Similarly, every genus found in South America does not occur in Africa, which simply is not true. Based on this figure, there are no pantropical/global genera in the input data, which is extremely strange and difficult to explain, considering that nearly 10% of global plant diversity is found amongst 22 large plant genera that include >1000 species each and are mostly global in their distribution. Scale is missing from Fig. S9, which is needed to understand the scale used in each subfigure. This needs to be revised so that the reader understands how many genera were not informative i.e. were found across all realms, and how many across different pairs of realms. Perhaps a ven-diagram would be a good way of showing this detail.

RESPONSE: Sorry for the unclear figure (i.e. Fig S9). Fig. S9 was intended to show the proportion of genera that are found in a certain floristic region in relation to all the genera

distributed in each geographic unit. However, the color scale of the original Fig S9 was not good to clearly demonstrate the patterns. In the revision, we updated the color scale of Fig S9, and included the legends for each map. Currently, the patterns in Fig S9 are much clearer. Following your suggestion, we calculated how many realms each genus is distributed in and showed the results in Appendix S5. We found 64% of all genera are endemic to one floristic realm, while only 50 (0.4%) genera are found to be distributed 5 realms and 2 (0.02%) in 6 realms. No genera were found to be distributed across over 7 realms. As you point out, some genera found in Australia also occur in other continents such as Asia and Africa, and some genera found in South America also occur in North America, Asia, and Africa (Appendix S5). We did not show the results as a ven-diagram because there are too many different combinations of realms.

Another aspect of the quality control would be the synonymy used for genera. I found several old synonyms in the dataset, e.g. *Lycopersicon* is a genus now nested in *Solanum*, but in the dataset of the paper, it was kept separate with a wide distribution which reflects cultivated species (tomato) which the authors argue they had dealt with having added methods removing non-natives. Other synonyms include *Symbegonia* (is now *Begonia*), etc. Please run your data through World Flora Online that get accepted generic names standardized would be my strong suggestion, if your current pipeline doesn't pick these obvious established synonyms at generic level. Plants of the World Online currently lists 13,973 accepted plant genera, which contrasts starkly against the number of 14,289 genera given in SI File page 7 methods section. The raw data should be ran against synonymy prior to all analyses on the distribution data. These synonymous names should not appear in the phylogeny used either.

RESPONSE: Thanks for your comments. We are sorry that the original appendix of distribution data (i.e. Appendix S2) mistakenly contained all the genera for which we have compiled distribution data, including some cultivated taxa and genera that are not matched in the phylogeny used in our study. Actually, only 12,778 angiosperm genera for which we have both distribution and phylogenetic data were used in our analysis (please see the Supplementary Methods, Page 7 of the SI file). The cultivated genera, such as *Lycopersicon*, are not among these 12,778 genera used in the analysis. In the revision, following your suggestions, we used the latest World Flora Online (WFO, <http://worldfloraonline.org/>) and Plants of the World Online

(POWO, <https://powo.science.kew.org/>) to check the synonyms of all genera. We also further checked and corrected some synonyms using the Taxonomic Name Resolution Service 4.0 (TNRS, <https://tnrs.biendata.org/>) which integrates data from the Tropicos (<https://www.tropicos.org>), United States Department of Agriculture (USDA), WFO, and World Checklist of Vascular Plants (WCVP). We found that there are 114 additional synonyms. We corrected these names in both the distribution and phylogeny data and re-ran the regionalization analysis. The results did not change because synonyms are less than 1% of the total genera and most of them have very narrow ranges. Please find the updated data in Appendix S2.

Figure 1 is great, but I think this would be best visualised as an unrooted dendrogram because you cannot argue to know the outgroup. The software generally roots on the most distant branch, but this assumption to make the most distant group an outgroup is not really needed in your work. An unrooted dendrogram would be a better way to visualise the similarity. Use FigTree software for example generally used for molecular phylogenetics.

RESPONSE: Thanks for your comments. Following your suggestion, we replaced the dendrogram with an unrooted dendrogram in Figure 1 and did the same for the supplementary Figures S2, 8 & 10.

Figure 2 would benefit from colour legend explaining the realms. What does pink refer to?

RESPONSE: Thanks for your comment. The floristic realms which can be matched to the present-day realms are shown in the same colors as shown in Fig. 1a. Due to the present-day floristic realms are not distinguishable in some historical periods, we used other colors to represent these ancestral floristic realms. Specifically, light green in maps of 10, 40 and 50 Ma represents the paleotropical realm covering the geographic ranges of the present-day African and Indo-Malesian realms; pink appearing from 100 Ma to 140 Ma represents the paleotropical realms covering the present-day Neotropical+African realms, the present-day Neotropical+African+Indo-Malesian realms, and the present-day Gondwanan super-realm, respectively. Interestingly, most present-day floristic realms are undistinguishable in 160 Ma. Please see the caption of Figure 2.

You state that only 19 families were found to be non-monophyletic: it would be important to list these somewhere at least in supplementary materials to understand which ones and to see the reasons for this. For example, knowing if it is Leguminosae and Asteraceae (two large families) due to overall polyphyly would be important.

RESPONSE: Thanks for your comments. We added the list of non-monophyletic family as Appendix S4. We also provided the number of genera of each of these families, and the information on whether the non-monophyly of these families is due to the inclusion of ITS. Both Leguminosae and Asteraceae mentioned by you are monophyletic in our phylogeny. Families that are not recovered as monophyletic are also families which generic composition is contentious (e.g., Capparaceae). Eighteen of the non-monophyletic families contain less than 25 genera within each of them and only one family (i.e. Plantaginaceae) contains more than 25 genera (i.e. 86 genera). Three of the non-monophyletic families, i.e., Capparaceae, Muntingiaceae, and Salvadoraceae, became monophyletic after excluding ITS. Please see Appendix S4 for details and see <https://en.geodata.pku.edu.cn/index.php?c=content&a=list&catid=199> (User: flowertree; Password: flowertree, which will be released after publication) for the phylogenetic topology

The revised methods state that the distribution data for each genus was checked manually. There were 12,778 genera in the final dataset, how was the manual check actually done? This seems a fantastic addition but is hard to understand how this was achieved. How many cases errors were spotted and corrected? (SI material page 6-7).

RESPONSE: Thanks for your comment. The genus-level distribution maps were verified by ecologists/taxonomists from several universities and institutes in China, including Peking University, Sichuan University, Institute of Botany of CAS, etc. We have been working on improving the data quality for both species and genus distributions in different regions continuously since ca. 2003, and these data are therefore the cumulative result from a long-term work, rather than a concentrated mission.

During this study, several professors, postdocs and students have looked through the genus distribution maps of different taxonomic groups from 2015 to the present. During the manual check of the distribution maps, new records were added, and dubious and cultivated records were removed according to the descriptions of the Plants of World Online

(<https://powo.science.kew.org/>) and many regional floras (such as *Flora of North America*, *Flora of China*, and *Flora of the USSR*).

We have included the updated genus distribution data used in this study in the supplementary materials of this manuscript and constructed a website to make the species distribution data available at: <https://en.geodata.pku.edu.cn/index.php?c=content&a=list&catid=198>.

We also included more descriptions about the quality of the distribution data. Specifically, the number of available data sources for different regions differed, leading to geographical variations in the confidence of the data quality. Based on the number of available data sources, we set different thresholds for the number of data sources to retain an occurrence record of a species in a geographical unit in different regions. For geographical units in Europe, Australia, China, South Africa, Madagascar and North America, an occurrence record of species in a geographical unit corroborated by at least 3 data sources was retained, leading to high confidence of the data quality in these regions. For the geographical units in Central America, Greenland, Amazon and Turkey, an occurrence record of species in a geographical unit corroborated by at least 2 data sources was retained, leading to medium confidence of the data quality in these regions. The entire data was retained for India, North and Central Africa, and Patagonia because of data deficiency in these regions, leading to relatively low confidence of the data quality in these regions. Please also see the reply to your comment below

You say you made distinct ITS alignments for orders. I see this is an advancement from previous general alignment of ITS across all angiosperms, but better option would be to chop the alignments into groups based on % divergence cut off. This is something for future work, I understand you have tried to do best possible approach.

RESPONSE: Thanks for your suggestions. We strongly agree with you. As you indicated, this important methodological issue deserves a future study. Since the phylogenies reconstructed with or without ITS in the current study are highly consistent and it would take a very long time to re-run these phylogenetic reconstructions, we would like to keep the results in the current study and explore this methodological issue in a future study on the reconstruction of angiosperm phylogeny following your suggestion.

The methods say that sources of species occurrence data, and it would be nice to see summary on data bias in terms of geography/realm/continent, or illustration of density of occurrence points used geographically in order to understand data quality.

RESPONSE: During data compilation, the number of available data sources for different regions differed, leading to geographical variations in the confidence of the data quality. Based on the number of available data sources, we set different thresholds for the number of data sources to retain an occurrence record of a species in a geographical unit in different regions (Fig. R1, below). For geographical units in Europe, Australia, China, South Africa, Madagascar and North America, an occurrence record of species in a geographical unit corroborated by at least 3 data sources was retained, leading to high confidence of the data quality in these regions. For the geographical units in Central America, Greenland, Amazon and Turkey, an occurrence record of species in a geographical unit corroborated by at least 2 data sources was retained, leading to medium confidence of the data quality in these regions. The entire data was retained for India, North and Central Africa, and Patagonia because of data deficiency in these regions, leading to relatively low confidence of the data quality in these regions.

Figure R1 The geographical variation in the confidence of the quality of genus distribution data.

Minor points:

The terms floristic and geographic are used, but sometimes as “geographical” (e.g., zoogeographical): I don’t know if there is a difference, but I would think better to use the term geographic (not geographical). This applies to biogeographic too (not biogeographical).

RESPONSE: Thanks. We replaced these words as suggested.

P. 3 line 38: remove word “regionalization”, not needed in my view

p. 3 line 41: remove “of the different regions”, not needed

p. 4 line 72: descendant (correct spelling)

p. 4 line 74: edit to “to the biogeographic regionalization due to ...”

p. 4 lines 75-76: edit to “and geographic distributions (34).”

Page 4 line 80: edit to “between regions at genus level.”

p. 5 first subheading: simpler heading “World’s floristic regionalization”

p. 5 line 108: add “due to climatic or geographic barriers”

p. 5 lines 109-115: add a note that these results “suggest recent exchange through dispersal of lineages between the two regions”.

p. 8 lines 209-214: first sentence of the paragraph is difficult, perhaps edit to simplify. Get rid of It’s and write out It is. Plate tectonics resulted in landmasses shifting in their climate but also climates kept changing globally over millions of years so clarify that both factors means that geographic isolation and climates may be linked.

p. 12 line 324: edit to “to be highly consistent”

RESPONSE: Done.

Reviewer #2 (Remarks to the Author):

The authors have invested a great deal of effort into their analyses, and I compliment them for their work.

RESPONSE: Thank you very much for your effort reviewing our manuscript again. We have revised the manuscript following your suggestions, and carefully addressed the uncertainties in floristic maps in the main text following your suggestions. We provided point-by-point responses to your comments as following.

The one important concern I raised previously was on the sensitivity of the identified realms to

the choice of clustering algorithm. The authors now include results using fuzzy c-means, and this is very useful. I do not completely agree with their justification for preferring UPGMA over the alternative, which seems to be based more on practical rather than statistical justifications (suppl lines 552-562), but this is a minor point. More importantly, the comparison of methods allows us to better evaluate confidences in the delineation of realms. For example, the separation of the Chile-Patagonian realm reported using phylo beta diversity (but not taxonomic diversity) is not recognised using fuzzy c-means, and thus appears to be particular to the clustering method used. This is an important observation, and should be clarified in the main text (perhaps at paragraph lines 109-115).

RESPONSE: Thanks for your comments. As suggested, we carefully addressed the inconsistencies between the realm boundaries based on the UPGMA and the fuzzy C-means methods in the main text, and discussed the possible reason for the inconsistency in the supplementary discussion. Please see lines 98-111 in the main text, and the supplementary discussion.

One key result, highlighted in the abstract, is the recognition of a “new” Saharo-Arabian realm. However, as we see in the silhouette analysis, this is also the realm that has a boundary that is less well defined, suggesting there might still be some uncertainty with respect to the distinctness of this floristic unit. The uncertainty in the delineation of this ‘realm’ is again demonstrated in the clustering by fuzzy c-means, which suggests that it might encroach into the European flora, I strongly recommend that the authors embrace this uncertainty in their description of this geographic unit (e.g. lines 165-173). We may well have evidence for a distinctive floristic region, but its bounds are still unclear.

RESPONSE: Thanks for your comments and we agree. We discussed the uncertainty on the boundary of the Saharo-Arabian realm in the main text as suggested. Please see lines 179-188. As you have pointed out, the separation of Saharo-Arabian realm from the Holarctic realm is robust to different assumptions on the crown age of angiosperms (Fig. S2), variations in phylogenetic topology (Fig. S3-7), sampling biases (Fig. S8), taxonomic beta diversity (Fig. S10) and the chosen of different clustering methods (Fig. S17). Even though, we removed the recognition of a “new” Saharo-Arabian realm from the abstract because of the uncertainty in its boundary.

The authors may find additional statements on the delineation of realms that are not fully supported by both methods, and I encourage them to make this clear to the reader in the main text, where appropriate. By recognising where there is uncertainty, I hope the paper will become stronger, and also highlight regions where floristic affinities are still unresolved.

RESPONSE: Thanks for your comments and we agree. We addressed in the main text that the Chile-Patagonian realm and the North American sub-realm as regions may have low confidence compared with the other realms due to the inconsistency between the results of UPGMA and the fuzzy c-means methods. We also addressed the uncertainty in the boundary of the Saharo-Arabian realm in the main text as you suggested. Please see lines 98-111, and lines 179-188 and the supplementary discussion.

Minor point, to avoid confusion, I suggest the authors refer to ‘present-day floras’ versus ‘extant floras’ in the section titled “The divergent times between the identified realms”. This is because we are looking at the ancestral affinities of the floras (extant) in the present day, not the floras that may have been extant in these regions historically.

RESPONSE: We agree, and we implemented these changes as suggested.

REVIEWER COMMENTS

Reviewer #1 (Remarks to the Author):

3rd Review of the re-submitted "An updated floristic map of the world"

The authors present an enhanced version of their previous submission that presents an updated floristic map of the world. The revisions are good, I have listed my questions regarding some of them. Many of the updated edits are fantastic, and add robustness and detail to the paper which is so merits. I thank the authors for taking time to respond in such detail. I really like your approach on how to use different thresholds of data sources (confidence) on different regions, fantastic step forward from current approaches seen in published papers on macroecological scale. However, I still have some questions about two aspects, which I explain in detail below.

Fig. S9: I agree this figure is now much enhanced and clearer. Just clarify: all genera found within a given realm (e.g., Chile-Patagonia in subfigure C) are shown throughout their range? If so, I would interpret the figure so that close to 100 of genera are endemic to the realm and not found outside, but meanwhile north America has light blue which would correspond to at least 40% indicating that 40% of the genera found in Chile-Patagonia also occur in North America. no matter how I try to interpret the explanation of the figure and the legend, I cannot read it "correctly" it seems without coming against issues. Perhaps what is illustrated is the % of genera in a flora of a given area (TDWG region in this case it seems) – again this explanation does not add up because the other subfigures illustrate up to 100% of genera found in the Neotropical realm are also found in Chile-Patagonia. This needs clarification. In your response letter you say that Fig S9 shows "the proportion of genera that are found in a certain floristic region in relation to all the genera distributed in each geographic unit." But I still don't understand how one area (e.g., Chile) can show near 100% in subfigure c and d in that case? Appendix S5 is fantastic and the presentation works. What I still disagree and find odd that none of the genera are distributed beyond 5 realms. This is simply not the case: genera such as Solanum is definitely distributed across all 9 realms, there is no question about that

(<https://powo.science.kew.org/taxon/urn:lsid:ipni.org:names:30000630-2>). There are 22 similarly large genera with >1000 species of which many show the same pattern, please check your data on these carefully. Of course, there are many more large genera with >500 but less than 1000 species, and some of them could also be globally distributed. Because of the large size of these genera, they are highly likely to be included in your phylogeny, and hence something is odd in that these genera do not show up in your results. This raises questions about some steps of the data gathering and analysis because how are you not picking up any genera that are distributed across 7 or more realms? Appendix S2 shows that Solanum is present in all major realms, so where does the error take place? Or am I reading S2 correctly?

Genera with >1000 species based on WFO:

Acacia

Anthurium

Astragalus <https://powo.science.kew.org/taxon/urn:lsid:ipni.org:names:330028-2>

Begonia

Bulbophyllum

Carex <https://powo.science.kew.org/taxon/urn:lsid:ipni.org:names:330029-2>

Croton

Dendrobium

Epidendrum

Eugenia

Euphorbia <https://powo.science.kew.org/taxon/urn:lsid:ipni.org:names:327729-2>

Impatiens <https://powo.science.kew.org/taxon/urn:lsid:ipni.org:names:325983-2>

Lepanthes

Miconia

Peperomia

Phyllanthus <https://powo.science.kew.org/taxon/urn:lsid:ipni.org:names:327609-2>

Piper

Psychotria

Rhododendron

Salvia <https://powo.science.kew.org/taxon/urn:lsid:ipni.org:names:30000096-2>

Senecio <https://powo.science.kew.org/taxon/urn:lsid:ipni.org:names:325904-2>

Solanum
Stelis
Syzygium

What are the 16 nested sub-realms (in terms of how are they supported by your data) you refer to in the abstract line 14? These are named in Table S1 but there is no support shown for these in any of the figures. I would like to see the unrooted dendrogram and a NMDS and a map (similar to Fig. 1) of these sub-realms in SI info. Or remove any focus on sub-realms.

In response to your reply on synonymy, synonyms such as *Lycopersicum* should not be removed but included within their current accepted genus name which is *Solanum*. Genus *Lycopersicum* includes wild species too, which are now part of the wider concept of *Solanum*. If you remove these records in your analysis, you reduce the geographic area assigned to a given accepted genus. This is not correct approach.

Minor comments and edits needed:

Main manuscript, line 224: change into "We then..."

Appendix S5: Please correct "Whether a genera are endemic to a certain realm or occupied across different realms" to read "Whether a genus is endemic to a certain realm or occupied across different realms"

SI File line 64: correct "non-monophonic" to say "non-monophyletic"

Line 65: correct "families which generic composition" to say "families in which generic composition"

Line 194: add detail that the 12,664 genera represent X[calculate]% of the total 13,974 accepted genera (POWO REF here).

Line 559: correct "These results indicate that these identification" to say "These results indicate that the identification"

Line 727: correct "credibility trees" to say "credibility tree"

Line 730: correct "the single maximum credibility trees" to say "the single maximum credibility tree"

Line 734: replace taxa with genera

Lines 755-757 (as well as in Fig. S10 legend): edit to read "The unrooted dendrogram depicts the relationships among floristic realms evaluated using UPGMA clustering method based on phylogenetic beta diversity between realms. The scale bar in the dendrogram shows the dissimilarity between realms."

Lines 760-762 (as well as in Fig. S10 legend): edit to read "Each tip in the dendrogram and each point in the scatter plot represents a geographic standard unit and the colors indicate the floristic realms that they belong to."

Line 765: POWO lists 13,974 accepted genera, and you have data for 12,664 (number in A S5, but 12,663 given in Fig. S9 legend!)

Line 785: should it be sub-realms or subrealms?

Fig. S12 and Fig. S13: here you refer to clusters of realms, but one of these is not supported by your data. For example, the Gondwanan and Laurasian superrealms reflecting north south divide in dendrogram presented in Fig. 1b in the main article is supported. The cluser of (Neotropical, Chile-Patagonian) realms is supported, as well as cluster of (African, Indo-Malesian), and a cluster of (Australian, Novozealandic). The supercluster of ((African, Indo-Malesian), (Australian, Novozealandic)) is not supported however. This affects subfigure c. I would suggest removing this comparison.

REPLIES TO REVIEWERS' COMMENTS

Reviewer #1 (Remarks to the Author):

Reviewer #1 (Remarks to the Author):

3rd Review of the re-submitted "An updated floristic map of the world"

The authors present an enhanced version of their previous submission that presents an updated floristic map of the world. The revisions are good, I have listed my questions regarding some of them. Many of the updated edits are fantastic, and add robustness and detail to the paper which is so merits. I thank the authors for taking time to respond in such detail. I really like your approach on how to use different thresholds of data sources (confidence) on different regions, fantastic step forward from current approaches seen in published papers on macroecological scale. However, I still have some questions about two aspects, which I explain in detail below.

RESPONSE: Thank you very much for your effort reviewing our manuscript again and for your positive comments. We apologize for the unclear Fig. S9. We updated its color scale to make it clearer (please see Figure R1). We also updated Figure 1 and Figure S2 to show the boundaries of the 16 sub-realms more clearly, and updated Appendix S5 following your suggestions. We added a new figure in the SI info to show the unrooted dendrogram and NMDS plot of the sub-realms. We provided point-by-point responses to your comments as following.

Fig. S9: I agree this figure is now much enhanced and clearer. Just clarify: all genera found within a given realm (e.g., Chile-Patagonia in subfigure C) are shown throughout their range? If so, I would interpret the figure so that close to 100 of genera are endemic to the realm and not found outside, but meanwhile north America has light blue which would correspond to at least 40% indicating that 40% of the genera found in Chile-Patagonia also occur in North America. no matter how I try to interpret the explanation of the figure and the legend, I cannot read it "correctly" it seems without coming against issues. Perhaps what is illustrated is the % of genera in a flora of a given area (TDWG region in this case it seems) – again this explanation does not add up because the other subfigures illustrate up to 100% of genera found in the Neotropical realm are also found in Chile-Patagonia. This needs clarification. In your response letter you say that Fig S9 shows "the proportion of genera that are found in a certain floristic region in relation to all the genera distributed in each geographic unit." But I still don't understand how one area (e.g., Chile) can show near 100% in subfigure c and d in that case?

RESPONSE: We are sorry that the unclear color of Figure S9 leads to misleading again. The minimum values of proportion in each subfigure is different, leading to variations in the color gradient among subfigures. Consequently, range of values represented by the darkest colors are not the same across subfigures. For example, the values of the genus proportions in the geographic units of the Patagonian realm differed between Figure S9(c) Figure S9(d), while their colors were indistinguishable in the original figure. To solve this issue, we improved the color gradient to make the colors comparable between different subfigures (please see Fig. R1).

Fig. R1 Natural ranges of 12,664 angiosperm genera. Color gradients indicate the proportions of genera in a geographic unit that are found in a certain floristic realm among all genera of that geographic

unit. **a)** Saharo-Arabian realm, **b)** Holarctic realm, **c)** Chile-Patagonian realm, **d)** Neotropical realm, **e)** Indo-Malesian realm, **f)** African realm, **g)** Australian realm, **h)** Novozealandic realm. See **Appendix S5** for detailed numbers of endemic genera within each floristic realm and the numbers of shared genera among different realms.

“Genera that are found in a certain floristic region” refer to the genera whose distributions cover a realm (including both endemic genera of the realm and wide-ranged genera whose distributions cover this realm, see Appendix S5 for the detailed number of endemic and wide-ranged genera in each realm). For each floristic realm, we extracted the list of genera that were found in the realm and then calculated the proportion of these genera to all co-existing genera in each geographic unit. We plotted the proportion values in Fig. R1 (i.e., the original Fig. S9) to reflect the division between floristic assemblages of different floristic realms. Notably, if a genus distributes across more than one floristic realm, it will be counted in the calculations for the realms that it occupies. Thus the sum of the proportion values in a geographic unit across all the subfigures can be over 100%. For example, a geographic unit in southern part of the Patagonian realm has proportion values of 100% and 85.6% in Figs. R1(c) and (d), respectively. This means that not all the genera of this geographic unit are endemic to the Chile-Patagonian realm. Instead, a large proportion of the species living in this geographic unit are also found in the Neotropical realm.

Fig. S9 seems to be too complicated for our understanding. Moreover, the information reflected in Fig. S9 is also reflected in Appendix S5, which is clear enough to show the number of endemic and wide-ranged genera in each floristic realm. And thus, we removed Fig. S9 from the manuscript and only kept Appendix S5 in order to avoid duplication and confusion.

Appendix S5 is fantastic and the presentation works. What I still disagree and find odd that none of the genera are distributed beyond 5 realms. This is simply not the case: genera such as *Solanum* is definitely distributed across all 9 realms, there is no question about that

(<https://powo.science.kew.org/taxon/urn:lsid:ipni.org:names:30000630-2>). There are 22 similarly large genera with >1000 species of which many show the same pattern, please check your data on these carefully. Of course, there are many more large genera with >500 but less than 1000 species, and some of them could also be globally distributed. Because of the large size of these genera, they are highly likely to be included in your phylogeny, and hence something is odd in that these genera do not show up in your results. This raises questions about some steps of the data gathering and analysis because how are you not picking up any genera that are distributed across 7 or more realms? Appendix S2 shows that *Solanum* is present in all major realms, so where does the error take place? Or am I reading S2 correctly?

Genera with >1000 species based on WFO:

Acacia

Anthurium

Astragalus <https://powo.science.kew.org/taxon/urn:lsid:ipni.org:names:330028-2>

Begonia

Bulbophyllum

Carex <https://powo.science.kew.org/taxon/urn:lsid:ipni.org:names:330029-2>

Croton

Dendrobium

Epidendrum

Eugenia

Euphorbia <https://powo.science.kew.org/taxon/urn:lsid:ipni.org:names:327729-2>

Impatiens <https://powo.science.kew.org/taxon/urn:lsid:ipni.org:names:325983-2>

Lepanthes

Miconia

Peperomia

Phyllanthus <https://powo.science.kew.org/taxon/urn:lsid:ipni.org:names:327609-2>

Piper

Psychotria

Rhododendron

Salvia <https://powo.science.kew.org/taxon/urn:lsid:ipni.org:names:30000096-2>

Senecio <https://powo.science.kew.org/taxon/urn:lsid:ipni.org:names:325904-2>

Solanum

Stelis

Syzygium

RESPONSE: In the previous version of Appendix S5, when counting the number of genera within a realm, a genus was identified as the genus of a particular realm only if >10% of its global distribution was found in that realm. As you pointed out, this approach will lead to biases, e.g., wide-ranged genera are sometimes missed when counting the number of genera distributed in each floristic realm, especially in those with small sizes (i.e., Novozealandic and Chile-Patagonian). Thus, in the updated Appendix S5, we removed the mentioned threshold and re-ran the same analysis. We found that 6,792 out of 12,664 (53.6%) genera are endemic in a certain realm and 87 (0.7%) genera are found across all realms. This mistake will only affect Appendix S5 and the other results of this manuscript did not change since all of the genera are included when conducting the analysis of this study. Please see the updated Appendix S5.

What are the 16 nested sub-realms (in terms of how are they supported by your data) you refer to in the abstract line 14? These are named in Table S1 but there is no support shown for these in any of the figures. I would like to see the unrooted dendrogram and a NMDS and a map (similar to Fig. 1) of these sub-realms in SI info. Or remove any focus on sub-realms.

RESPONSE: We used dashed lines to show the boundaries of sub-realms in the Figure 1 and Figure S2, which was not clear enough. As suggested, we updated the boundaries of sub-realms in Figure 1 and Figure S2 to make them clearer (please see Figure 1 and Figure S2 in the revised version). Moreover, we also added a new figure to show the unrooted dendrogram, a NMDS and a map of sub-realms (similar to Fig. 1) as suggested (please see Fig. S1).

In response to your reply on synonymy, synonyms such as *Lycopersicum* should not be removed but included within their current accepted genus name which is *Solanum*. Genus *Lycopersicum* includes wild species too, which are now part of the wider concept of *Solanum*. If you remove these records in your analysis, you reduce the geographic area assigned to a given accepted genus. This is not correct approach.

RESPONSE: We apologized for the misleading statement in the second round of our response. When we checked the synonyms of all genera in the last round, we included the species of the synonym genera into their current accepted genera accordingly. To eliminate the influence of introduced species and distributions, we checked the statuses of each species of the synonym general, and whether distributions of these species are native according the Plants of the World Online (<https://powo.science.kew.org/>) and efloras (<http://www.efloras.org/>, accessed: May, 2019). We only kept the wild species and their natural distributions in our database when we merged the synonym genera with their accepted genera. Let's take species in *Lycopersicum* as an example. All wild species in *Lycopersicum* and their natural distributions were included into its current accepted genera *Solanum*. But the cultivated records of *Lycopersicum* were excluded. This process was also used for other synonym genera. We checked and updated the method again to make it clear.

Minor comments and edits needed:

Main manuscript, line 224: change into “We then...”

Appendix S5: Please correct “Whether a genera are endemic to a certain realm or occupied across different realms” to read “Whether a genus is endemic to a certain realm or occupied across different realms”

SI File line 64: correct “non-monophonic” to say “non-monophyletic”

Line 65: correct “families which generic composition” to say “families in which generic composition”

Line 194: add detail that the 12,664 genera represent X[calculate]% of the total 13,974 accepted genera (POWO REF here).

Line 559: correct “These results indicate that these identification” to say “These results indicate that the identification”

Line 727: correct “credibility trees” to say “credibility tree”

Line 730: correct “the single maximum credibility trees” to say “the single maximum credibility tree”

Line 734: replace taxa with genera

Lines 755-757 (as well as in Fig. S10 legend): edit to read “The unrooted dendrogram depicts the relationships among floristic realms evaluated using UPGMA clustering method based on phylogenetic beta diversity between realms. The scale bar in the dendrogram shows the dissimilarity between realms.”

Lines 760-762 (as well as in Fig. S10 legend): edit to read “Each tip in the dendrogram and each point in the scatter plot represents a geographic standard unit and the colors indicate the floristic realms that they belong to.”

RESPONSE: Thank you for providing these writing suggestions. All the above were done as suggested.

Line 765: POWO lists 13,974 accepted genera, and you have data for 12,664 (number in A S5, but 12,663 given in Fig. S9 legend!)

RESPONSE: Yes, we have 12,664 rather than 12,663. Sorry for the mistake, and we have updated it.

Line 785: should it be sub-realms or subrealms?

RESPONSE: We checked and use sub-realms throughout the manuscript.

Fig. S12 and Fig. S13: here you refer to clusters of realms, but one of these is not supported by your data. For example, the Gondwanan and Laurasian superrealms reflecting north south divide in dendrogram presented in Fig. 1b in the main article is supported. The cluster of (Neotropical, Chile-Patagonian) realms is supported, as well as cluster of (African, Indo-Malesian), and a cluster of (Australian, Novozealandic). The supercluster of ((African, Indo-Malesian), (Australian, Novozealandic)) is not supported however. This affects subfigure c. I would suggest removing this comparison.

RESPONSE: Thank you for pointing it out. The supercluster of ((African, Indo-Malesian), (Australian, Novozealandic)) is not supported, instead, supercluster of ((African, Indo-Malesian), (Neotropical, Chile-Patagonian)) is supported. Thus, we replaced this comparison in Figs.4-5 and Figs. S12-13 as suggested.

REVIEWERS' COMMENTS

Reviewer #1 (Remarks to the Author):

Thank you so much for clear explanations and revisions to the points I raised. I feel all of my questions and concerns have been addressed, and the manuscript is in a much enhanced state. I would include the figure R1 as it now stands somewhere in the supplementary information file, it is a great representation of your data and helpful in understanding the distribution patterns. I added a few tracked changes to the supplementary information file.

REPLIES TO REVIEWERS' COMMENTS

Reviewer #1 (Remarks to the Author):

Thank you so much for clear explanations and revisions to the points I raised. I feel all of my questions and concerns have been addressed, and the manuscript is in a much enhanced state. I would include the figure R1 as it now stands somewhere in the supplementary information file, it is a great representation of your data and helpful in understanding the distribution patterns. I added a few tracked changes to the supplementary information file.

Response: Thank you for your efforts on reviewing our paper. As suggested, we included the figure R1 as Fig. S19 as suggested and accepted your changes on grammar in the supplementary information file.